# Nelfb promotes dermal white adipose tissue formation through RNA polymerase II-mediated adipogenic gene regulation

Samiksha Mahapatra, Julian Gomez, Uyanga Batzorig, Ye Liu, Celia Fernández-Méndez, Yifang Chen and George L. Sen*

## ABSTRACT

Dermal white adipose tissue (dWAT) is crucial for skin homeostasis, contributing to hair follicle regeneration, immune defense and skin wound healing. dWAT is formed and maintained by the differentiation of adipocyte precursors found in the dermis of the skin. While transcription factors that control adipocyte differentiation have been well characterized, other aspects of transcription control, such as pausing/elongation, are poorly understood. Here, we show that deletion of the transcriptional pause factor, Nelfb, from preadipocyte lineages in mice led to a failure of dWAT and other fat depot formation, perinatal lethality and reduced expression of adipogenic genes. Nelfb promotes an open chromatin structure and stabilizes RNA Polymerase II binding to *Pparg*, *Cebpa*, *Krox20* and *Stat3* to allow their transcription, which is necessary for adipocyte differentiation. Retroviral expression of *Pparg* in Nelfb-depleted cells restored adipocyte differentiation in cultured cells, while treatment of Nelfb-deleted mice with the Pparg agonist, rosiglitazone, allowed for dWAT formation and prolonged lifespan. These findings highlight the essential role of Nelfb in promoting the expression of key adipogenic genes that are necessary for dWAT formation and adipocyte differentiation.

KEY WORDS: Dermal white adipose, Adipocyte, Adipogenesis, *Nelfb*, *Pparg*, Transcription elongation, Cebp, Sox9, Wnt signaling, Transcriptional pausing

## INTRODUCTION

Dermal white adipose tissue (dWAT), once seen as a simple fat depot, is now recognized as a dynamic layer essential for skin homeostasis (Zwick et al., 2018; Guerrero-Juarez and Plikus, 2018). dWAT is a significant contributor to the thickness of the skin, which can dramatically expand and shrink due to hair follicle cycling, cold stress or response to bacterial infection (Festa et al., 2011; Kasza et al., 2014; Zhang et al., 2015). This is in contrast to the more widely studied subcutaneous or visceral white adipose tissues, which have a relatively slower turnover rate (Spalding et al., 2008; Arner et al., 2011; Wang et al., 2013). Intradermal adipocytes are necessary for hair follicle regeneration through PDGF signaling

and to secrete antimicrobial peptides such as cathelicidin during *Staphylococcus aureus*-mediated infection of the skin (Zhang et al., 2015; Festa et al., 2011). In addition, dWAT is necessary for skin wound repair through adipocyte lipolysis and subsequent lipid release (Shook et al., 2020; Zhang et al., 2021). This in turn causes recruitment of macrophages and generation of extracellular matrix from fibroblasts. Given the importance of dWAT for promoting healthy skin, it is necessary to understand how dWAT develops and is maintained. During development of the skin, reticular fibroblasts of the dermis give rise to preadipocytes, which form dWAT (Driskell et al., 2013). This population of adipocyte stem/progenitor cells undergoes self-renewal through Pdgfa/Pdgfra signaling, which maintains dWAT mass (Rivera-Gonzalez et al., 2016). Studies on how the preadipocyte fate is controlled as well as how they differentiate into mature white adipose tissue has mainly focused on transcription factors. This includes identification of DLK1 (Kim et al., 2007), SOX9 (Gulyaeva et al., 2018), MEIS1 (Gulyaeva et al., 2018) and PBX1 (Monteiro et al., 2011), which promote the precursor fate, while transcription factors such as PPARG, CEBPA, CEBPB, KROX20 (also known as EGR2) and STAT3 drive differentiation towards adipocytes (Lee et al., 2019). A potentially potent but understudied way to regulate adipogenic fate may include transcriptional pause/elongation mechanisms. We have already demonstrated the importance of this type of regulation in the epidermis, where transcription elongation factors such as SPT6 (SUPT6; Li et al., 2021) and CDK12 (Li et al., 2022) are necessary for promoting epidermal differentiation, while ELL (Li et al., 2020) is necessary for maintaining the epidermal progenitor state. Promoter-proximal pausing and subsequent release of RNA Polymerase II (RNA Pol II) is a key regulatory mechanism in metazoans, enabling rapid gene expression responses to environmental and developmental cues (Smith and Shilatifard, 2013). RNA Pol II pausing could potentially be a method to generate a poised state during development in which crucial gene promoters are loaded with paused RNA Pol II in anticipation of future activation (Liu et al., 2015). Negative elongation factor (NELF: composed of NELFA, NELFB, NELFC/D and NELFE), in collaboration with DRB sensitivity-inducing factor [DSIF: SPT4 (SUPT4) and SPT5 (SUPT5)], promotes RNA Pol II pausing (Vos et al., 2018). RNA Pol II pausing is released once positive elongation factor b (P-TEFb: CDK9 and cyclin T) phosphorylates DSIF, NELF and serine 2 (Ser2) of the RNA Pol II C-terminal domain (CTD) (Saunders et al., 2006). Upon phosphorylation, NELF dissociates from RNA Pol II, which allows RNA Pol II and DSIF to facilitate transcription elongation. *In-vivo*, Nelfb (a key component of NELF) is required for embryogenesis (Amleh et al., 2009). The role of Nelfb is dependent on the type of tissue or cells targeted, making it important to elucidate its function in different tissues. In certain contexts, Nelfb is important for proliferation,

Department of Dermatology, Department of Cellular and Molecular Medicine, Division of Epithelial Biology, University of California, San Diego, La Jolla, CA 92093-0869, USA.

*Author for correspondence (gsen@health.ucsd.edu)

G.L.S., 0000-0003-1279-8550

while in others it is required for proper functioning of a tissue. For example, Nelfb is necessary for embryonic stem cell proliferation (Williams et al., 2015) and expansion of muscle progenitor cells due to injury (Robinson et al., 2021). In the endometrium, Nelfb is needed for hormonal signaling (Hewitt et al., 2019) and it is essential for maintaining metabolic homeostasis in cardiomyocytes (Pan et al., 2014). Nelfb has also been shown to be necessary for mammary gland development (Nair et al., 2016) and T cell function (Wu et al., 2022). Notably, the role of Nelfb and RNA Pol II pausing in adipogenic fate determination is unclear. Here, we show that loss of Nelfb in preadipocyte lineages results in the absence of dWAT as well as other fat depot formation and perinatal lethality. Nelfb promotes RNA Pol II stabilization to the transcription start sites (TSS) of *Pparg*, *Cebpa*, *Krox20* and *Stat3* in adipocyte precursors, which primes the cells for an adipogenic fate. Without Nelfb, RNA Pol II destabilizes and chromatin condenses causing these regulators to be repressed, which prevents adipocyte differentiation. Retroviral *Pparg* expression restores adipocyte differentiation in Nelfb-deficient cells. Furthermore, rosiglitazone (Rosi) treatment of Nelfb-deficient mice restores dWAT formation and extends lifespan, highlighting the crucial role of Nelfb in adipogenesis.

## RESULTS

### Nelfb$^{-/-}$ mice exhibit a disrupted dermal fat layer with a loss of dermal adipocytes

To determine if transcription elongation has any role in dWAT development or maintenance, we deleted Nelfb in preadipocyte lineages by crossing Nelfb$^{flox/flox}$ mice with Pdgfra-Cre$^+$ mice (Rivera-Gonzalez et al., 2016; Jeffery et al., 2014). This targeted knockout led to a marked reduction in *Nelfb* gene expression in the dermis of Nelfb$^{-/-}$ (Nelfb$^{fl/fl}$ Pdgfra-Cre$^+$) mice compared to controls (CTL: Nelfb$^{wt/wt}$ Pdgfra-Cre$^+$ or Nelfb$^{fl/wt}$ Pdgfra-Cre$^+$) (Fig. 1A,B). Nelfb$^{-/-}$ mice were significantly smaller and lower in body weight at postnatal day (P)6 (Fig. 1C,D). We analyzed skin morphology using Hematoxylin and Eosin (H&E) staining, which revealed that Nelfb$^{-/-}$ dorsal skin was thinner than control skin (Fig. 1E-G) at P6. The thinner skin was due to the absence of formation of the dWAT layer, the skin being devoid of dermal adipocytes, which are normally interspersed between hair follicles within the dermis (Fig. 1E,F). To validate the absence of mature adipocytes in the Nelfb-deleted mice, Oil Red O staining was performed to visualize lipid-filled adipocytes in the dermis. Nelfb$^{-/-}$ dermis showed drastically reduced Oil Red O-positive staining, in contrast to the CTL dermis (Fig. 1H,I). Notably Pparg (Rosen et al., 1999), a master regulator of adipogenesis, was decreased in the Nelfb$^{-/-}$ dermis compared to controls (Fig. 1J,K). On the mRNA level, Nelfb deletion led to the downregulation of key adipogenic enzymatic, metabolic and regulator genes such as *Acot4*, *Adig*, *Plaat3*, *Elovl6*, *Pparg*, *Cebpa*, *Plin1*, *Krox20*, *Adipoq* and *Fabp4* from the dermis (Fig. 1L). The majority of the Nelfb$^{-/-}$ mice died perinatally by P9 (Fig. 5B, Fig S1). However, to determine if there was delayed development of dWAT, we harvested mice that had survived until P9. By P9, the difference in size and weight between control and Nelfb-deleted mice was even more pronounced (Fig. S1A,B). Nelfb-deleted skin was thinner and lacked dWAT formation, suggesting a permanent loss of adipocytes from the dermis (Fig. S1C-G). Adipogenic regulators and markers such as *Pparg*, *Plin1*, *Adipoq* and *Fabp4* were significantly downregulated from the dermis harvested from Nelfb$^{-/-}$ mice (Fig. S1H). On the protein level, Perilipin 1 (Plin1), a marker for mature adipocytes, was downregulated in the dermis of Nelfb-depleted mice (Fig. S1I,J). The loss of dWAT and mature adipocytes could result from an absence or reduction of adipocyte progenitor

cells or an inability of these cells to differentiate due to loss of Nelfb. To determine if there were any changes to the number of adipocyte precursor cells, we used flow cytometry to quantify the percentage of Sca1 (Ly6A)$^+$ PDGFRA$^+$ preadipocytes (Zhang et al., 2015) from the dorsal skin of control and Nelfb$^{-/-}$ mice. Loss of Nelfb had no significant impact on the percentage of preadipocytes in the dermis, suggesting that the failure to develop dWAT was not due to an absence of precursor cells (Fig. S1K,L). This suggests that, although the progenitor population remains unchanged, they may have impaired ability to differentiate into functional dermal adipocytes.

### Nelfb is necessary for formation of white and brown adipose tissue

As the Nelfb$^{-/-}$ mice weighed much less than controls and Pdgfra-driven Cre-mediated deletion of Nelfb would occur in adipocyte progenitors of both brown and white fat depots (Jeffery et al., 2014; Lee et al., 2012), we wanted to determine whether Nelfb is also necessary for their formation. The weights of interscapular brown adipose tissue (BAT), inguinal white adipose tissue (iWAT) and mesenteric visceral white adipose tissue (vWAT) were significantly reduced in Nelfb-deleted mice (Fig. S1M). Similar to dWAT, these other adipose tissue depots showed loss of mature adipocytes as they had diminished Oil Red O staining (Fig. S1N,O). These results suggest the importance of Nelfb in adipogenesis in different adipose compartments in the body.

### Nelfb is required for expression of adipogenic transcription factors to induce adipocyte differentiation

To test whether the Nelfb-deleted adipocyte precursor cells could no longer differentiate into mature adipocytes, we isolated fibroblasts from newborn (P0) CTL and Nelfb$^{-/-}$ mice through enzymatic digestion of the dorsal dermis (Fig. S2A). Subsequently, we induced adipocyte differentiation by placing the cells in adipocyte differentiation medium for 14 days. After 14 days, Oil Red O staining revealed lipid accumulation in CTL adipocytes, while Nelfb-depleted cells showed no staining, indicating impaired adipocyte differentiation (Fig. 2A). Additionally, Pparg and Plin1 protein levels were present in control adipocytes but significantly reduced in Nelfb$^{-/-}$ cells (Fig. 2B, Fig. S2B-D). RNA-sequencing (RNA-seq) was also performed on CTL and Nelfb-depleted cells cultured in adipocyte differentiation medium for 14 days to comprehensively examine the gene expression changes due to loss of Nelfb. This analysis aimed to uncover the molecular pathways and regulatory networks controlled by Nelfb during adipocyte differentiation. In total, 1120 genes [false discovery rate (FDR)≤0.05 and ≥2-fold change] were differentially expressed upon Nelfb loss (Fig. 2C, Table S1). A total of 358 downregulated genes were found to be significantly enriched for Gene Ontology (GO) terms associated with processes such as fat cell differentiation and adipose tissue development, further emphasizing the important role of Nelfb in driving the differentiation of adipocytes (Fig. 2D,E, Table S1). We found that 23 key adipogenic genes (i.e. *Bckdha*, *Aldh6a1*, *Echs1*, *Bckdhb*, *Dbt*, *HSD17B10*, *Bcat2*, *Acadl*, *Gpx4*, *Gstp1*, *Elovl3*, *Acot2*, *Hacl1*, *Acot4*, *Plaat3*, *Sorl1*, *Ffar4*, *Adig*, *Adipoq*, *Atf5*, *Rarres2*, *Zbtb16*, *Trib3*) from the top four downregulated GO terms were decreased in Nelfb$^{-/-}$ cells (Fig. 2E). Validation of the RNA-seq by RT-qPCR further confirmed a marked downregulation of adipogenic genes such as *Adipoq*, *Plin1*, *Krox20*, *Cebpb*, *Mlxipl*, *Ebf1* and *Fabp4* in Nelfb-deleted cells (Fig. S2E). Furthermore, KEGG Pathway Analysis of the downregulated genes showed that there was a loss in the PPARG signaling pathway, along with other processes such as fatty acid elongation and unsaturated fatty acid biosynthesis

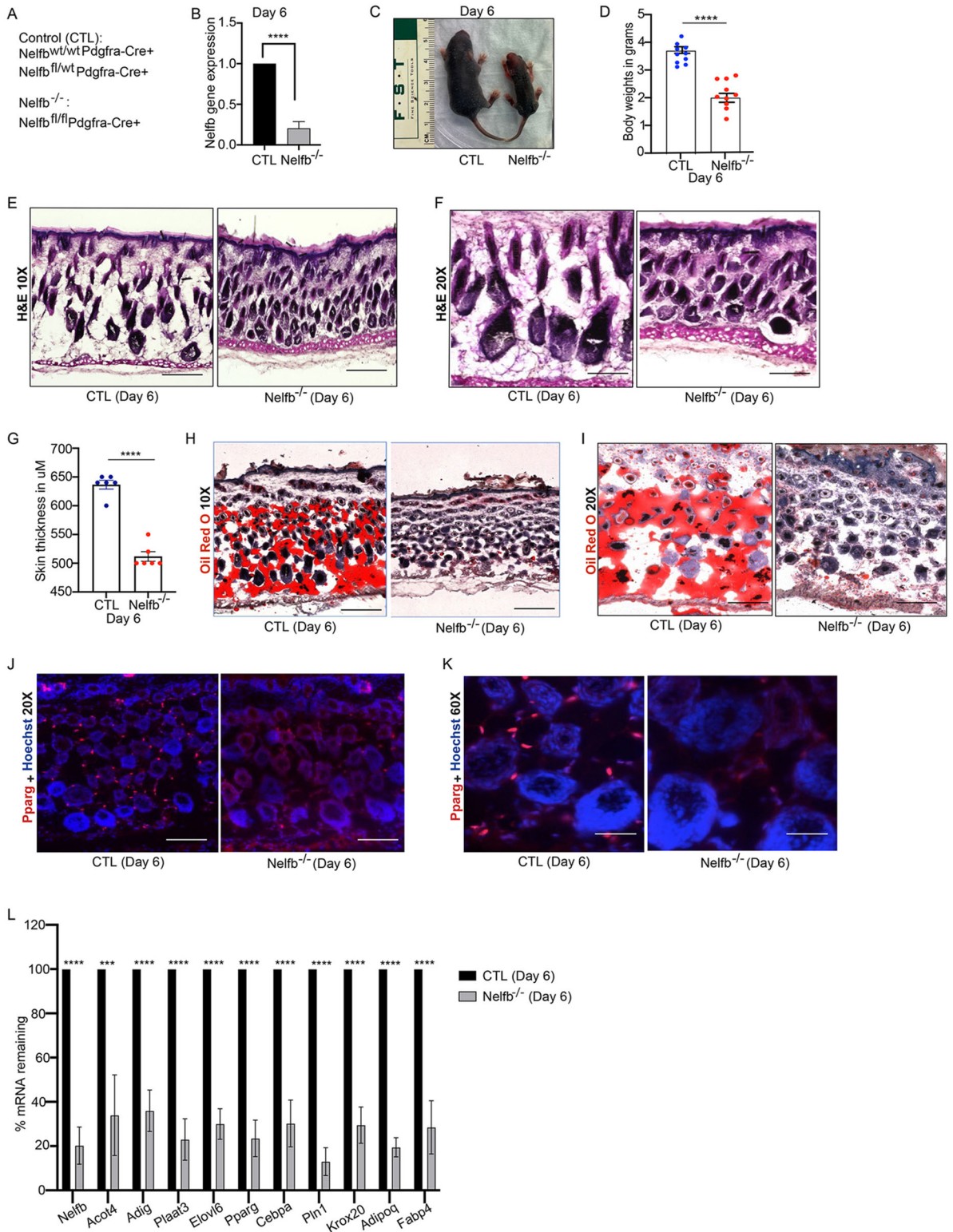

**Fig. 1. Nelfb is necessary for dWAT formation in the dorsal skin.** (A) Genotypes of control (CTL: *Nelfb^fl/wt^ Pdgfra-Cre^+^* or *Nelfb^wt/wt^ Pdgfra-Cre^+^*) and *Nelfb^−/−^* (Nelfb^fl/fl^ Pdgfra-Cre^+^) mice that were used in this study. (B) RT-qPCR of *Nelfb* gene expression from CTL and *Nelfb^−/−^* mice at P6. RT-qPCR was performed on the dermis that was isolated from dorsal skin (*N*=4). (C) Representative image of CTL and *Nelfb^−/−^* mice at P6. (D) Body weights of CTL and *Nelfb^−/−^* mice at P6. All individual dots in bar graphs represent data from an individual mouse. (E,F) H&E staining of CTL and *Nelfb^−/−^* dorsal skin harvested at P6 at 10× (E) and 20× (F) magnification (*N*=6). (G) Quantification of the dorsal skin thickness at P6. (H,I) Oil Red O staining of CTL and *Nelfb^−/−^* dorsal skin harvested at P6 at 10× (H) and 20× (I) magnification (*N*=6). (J,K) Staining of dorsal skin of CTL and *Nelfb^−/−^* mice with antibodies against Pparg (adipocyte marker: red) and Hoechst 33342 (nuclei stain: blue) at 20× (J) and 60× (K) magnification (*N*=4). (L) RT-qPCR of adipogenic genes on CTL and *Nelfb^−/−^* dermis at P6 (*N*=4). Results were normalized to housekeeping gene, *Gapdh*. Data are mean±s.e.m. ***P<0.01, ****P<0.0001 (two-tailed unpaired *t*-tests). Scale bars: 300 µm (E,H); 150 µm (F,I,J); 50 µm (K).

(Fig. S2F). The 762 upregulated genes were found to be enriched for GO terms involved in regulation of the Wnt pathway and positive regulation of interleukin 6 production (Fig. 2D, Fig. S2G, Table S1). The Wnt pathway inhibits adipogenesis by suppressing adipogenic factors, including Pparg and Cebpa, which are crucial for preadipocyte differentiation (Christodoulides et al., 2009). Importantly, the upregulated genes were enriched in factors such as *Dlk1* (Kim et al., 2007), *Wnt10b* (Longo et al., 2004), *Sox9* (Gulyaeva et al., 2018), *Meis1* (Gulyaeva et al., 2018) and *Pbx1* (Monteiro et al., 2011), which are known preadipocyte markers that act by maintaining the preadipocyte state and preventing adipocyte differentiation (Fig. S2H, Table S1). Given the inhibition of adipocyte differentiation observed with Nelfb depletion, it was important to assess whether this effect is linked to altered cell proliferation. To address this, CTL and Nelfb$^{-/-}$ cells cultured in adipocyte differentiation medium for 14 days were stained with antibodies against Ki67 to quantify the proportion of proliferating cells. There was a ~40% increase in Ki67$^+$ nuclei in Nelfb-deleted cells compared to control adipocytes, indicating that Nelfb-depleted cells remain proliferative but fail to differentiate into functional adipocytes (Fig. S2I,J). These results suggest that Nelfb is required for adipocyte differentiation and, in its absence, the cells are locked in a preadipocyte stage that are actively proliferating.

### Nelfb promotes open chromatin and RNA Pol II stability at adipogenic genes

To gain an understanding of whether Nelfb directly regulates adipogenesis genes, we performed CUT&RUN using an antibody targeting Nelfb to determine its bound genes. In CTL adipocytes, Nelfb bound strongly to the TSS of key adipogenic transcription factors, including *Pparg*, *Cebpa*, *Stat3* and *Krox20* (Fig. 2F). This binding, however, was lost in Nelfb$^{-/-}$ cells, demonstrating the specificity of the binding (Fig. 2F). Nelfb can regulate transcription by promoting RNA Pol II stabilization, pausing or elongation (Su and Vos, 2024). To test how loss of Nelfb impacts RNA Pol II binding on the genes coding for adipogenic transcription factors, an RNA Pol II CUT&RUN was performed. RNA Pol II showed robust binding to the TSS and in the gene body of these transcription factors in CTL adipocytes (Fig. 2G,H). In contrast, RNA Pol II binding to the TSS and gene body regions of *Pparg*, *Cebpa*, *Stat3* and *Krox20* was significantly diminished in Nelfb$^{-/-}$ cells (Fig. 2G,H). These findings suggest that Nelfb plays a vital role in maintaining the transcriptional activity of key adipogenic genes by stabilizing RNA Pol II binding at those genes. To see if Nelfb was directly influencing transcription elongation, CUT&RUN using an antibody against elongating forms of RNA Pol II (RNA Pol II Ser2 phosphorylation) was carried out. Robust RNA Pol II Ser2 phosphorylation could be detected on the TSS and gene body regions of *Pparg*, *Cebpa*, *Stat3* and *Krox20*, which disappeared upon deletion of Nelfb (Fig. 2I,J). This suggests that Nelfb is necessary for the elongation of these genes during adipogenic differentiation. To investigate how loss of Nelfb and RNA Pol II binding to these adipogenic transcription factors affects chromatin accessibility, CUT&RUN was performed using markers of chromatin accessibility and closure (Bannister and Kouzarides, 2011). H3K27Ac (open/active chromatin) and H3K4me3 (active transcription) were highly enriched at the TSS region of adipogenic transcription factors, but lost in Nelfb-deleted cells (Fig. 2K, Fig. S3A). In CTL adipocytes, there was also significant H3K27Ac enrichment in the gene body of key adipogenic transcription factors, but this was decreased in Nelfb-depleted cells (Fig. S3B). We also examined the repressive chromatin marks H3K9me3 (heterochromatin) and H3K27me3

(Polycomb repression) and found a substantial increase in their enrichment at both the TSS and the gene body of adipogenic transcription factors in Nelfb$^{-/-}$ cells compared to CTL adipocytes (Fig. S3C-F). These findings highlight the role of Nelfb role in maintaining active chromatin states essential for the transcription of adipogenic genes. Upon Nelfb loss, there is widespread chromatin closure at adipogenic genes, which suppresses adipogenesis. Next, we wanted to explore the effects of Nelfb on RNA Pol II stability on adipocyte precursor genes such as *Dlk1*, *Sox9*, *Pbx1* and *Meis1* since these genes were upregulated in Nelfb$^{-/-}$ cells (Fig. S2H). In CTL adipocytes, Nelfb and RNA Pol II did not bind to any of the progenitor genes whereas RNA Pol II was significantly bound in the Nelfb deleted cells (Fig. S3G,H). This correlates with the increased gene expression of *Dlk1*, *Sox9*, *Pbx1* and *Meis1* in the Nelfb-deleted cells (Fig. S2H). Since these experiments were performed in differentiated adipocytes, they did not address whether Nelfb binds and is required for RNA Pol II stabilization on the progenitor genes in undifferentiated cells, which is where the genes are expressed. To evaluate this, CUT&RUN was performed on CTL and Nelfb$^{-/-}$ cells cultured in growth medium rather than adipocyte differentiation medium. Nelfb bound to *Dlk1*, *Sox9*, *Pbx1* and *Meis1* in adipocyte precursor CTL cells, whereas the binding was significantly reduced in Nelfb-deleted cells (Fig. S4A). Interestingly, RNA Pol II bound more robustly to the progenitor genes in Nelfb$^{-/-}$ cells, suggesting that Nelfb is not required for RNA Pol II stabilization at the TSS of *Dlk1*, *Sox9*, *Pbx1* and *Meis1* (Fig. S4B). This suggests that Nelfb is required for stabilization of RNA Pol II at differentiation but not progenitor genes. Supporting this, Nelfb was already bound to the TSS of adipogenic genes (*Pparg*, *Cebpa* and *Krox20*) in CTL progenitor cells (Fig. S4C). This binding was correlated with stabilization of RNA Pol II at the TSS, as deletion of Nelfb results in loss of RNA Pol II from the adipogenic genes (Fig. S4D). Notably, there was only weak binding of RNA Pol II to the gene body of adipogenic genes in CTL progenitor cells, which is further diminished in Nelfb$^{-/-}$ cells (Fig. S4E). There is a lack of strong binding of RNA Pol II to the gene body of adipogenic genes until the CTL cells are differentiated in adipogenic medium for 14 days, suggesting that RNA Pol II is primarily paused at the TSS region in progenitor cells (Fig. S4E). As Nelfb is already bound to *Pparg*, *Cebpa* and *Krox20* in progenitor cells, we wanted to determine if these genes were enriched for any transcription factor motifs that could potentially recruit Nelfb to these sites (Fig. S4C). A search of the promoter region using TRANSFAC (Wingender et al., 2000) (−500 bp to TSS) of *Pparg*, *Cebpa*, *Stat3* and *Krox20* showed that only the SP1 binding motif was found in all four genes (Fig. S4F). Sp1 has been previously shown to bind and repress *Cebpa* in adipocyte progenitor cells (Tang et al., 1999). Sp1 may potentially repress these genes by recruiting Nelfb to these sites and promoting RNA Pol II pausing and thereby preventing expression of these genes in preadipocytes (Fig. S4E). These results suggest that Nelfb promotes paused RNA Pol II on adipogenic genes to prime progenitor cells for an adipogenic fate.

### Retroviral expression of *Pparg* restores adipocyte differentiation in Nelfb$^{-/-}$ cells

As Nelfb binds and is necessary for the stability of RNA Pol II-mediated transcription of *Pparg*, we hypothesized that exogenous expression of *Pparg* may restore adipocyte differentiation in Nelfb$^{-/-}$ cells. To do this, control and Nelfb$^{-/-}$ fibroblasts were harvested from the dorsal skin of newborn pups and infected with the pBabe retrovirus expressing *Pparg*. We evaluated three retroviral

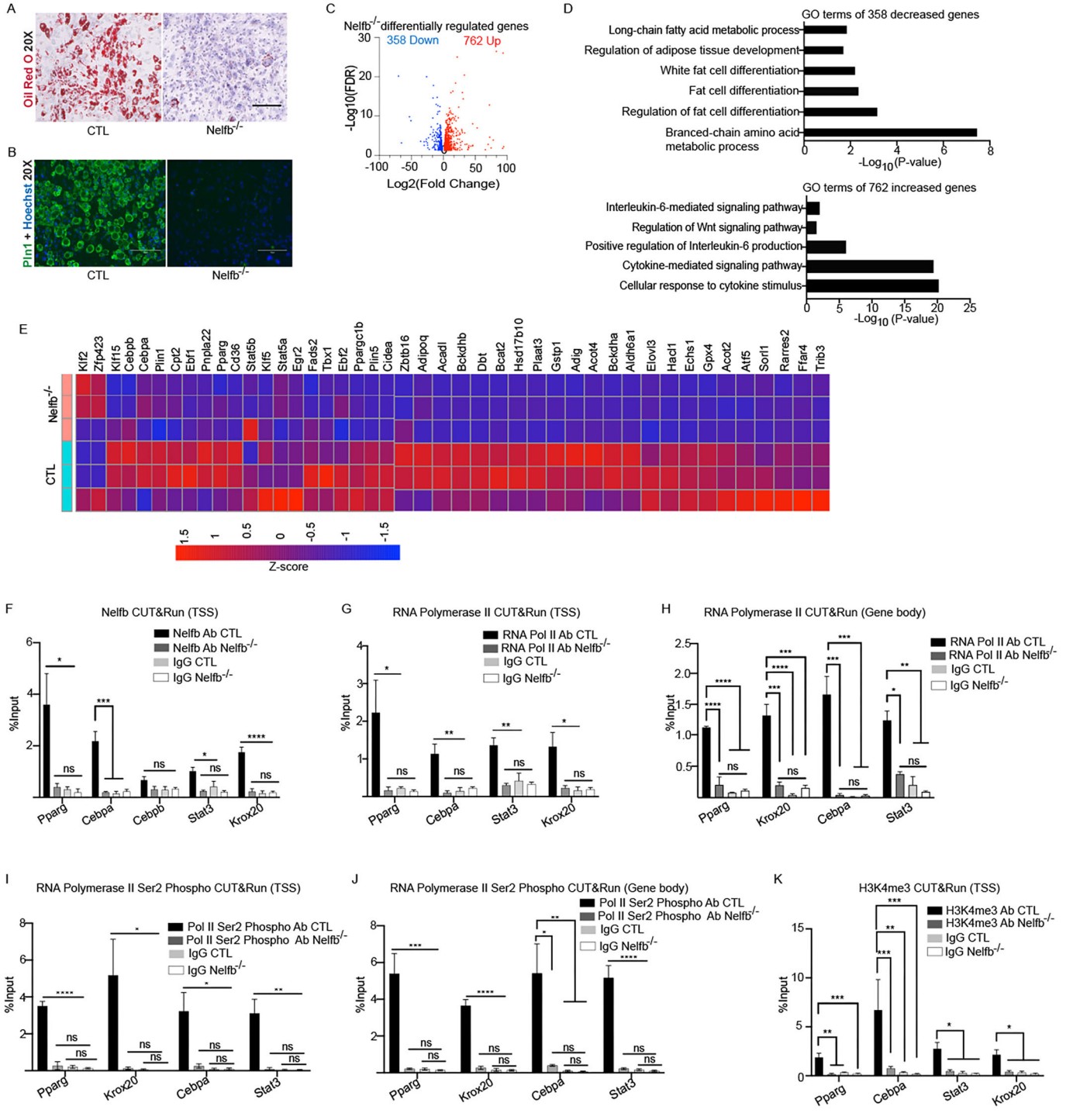

**Fig. 2. Nelfb is necessary for adipocyte differentiation by promoting the expression of adipocyte transcription factors.** (A) Oil Red O staining of control (CTL: *Nelfb*^fl/wt *Pdgfra-Cre*^+ or *Nelfb*^wt/wt *Pdgfra-Cre*^+) and *Nelfb*^−/− (*Nelfb*^fl/fl *Pdgfra-Cre*^+) cells cultured in adipocyte differentiation medium for 14 days (*N*=4). (B) Staining of CTL and *Nelfb*^−/− cells with antibodies against perilipin 1 (Plin1; mature adipocyte marker: green) and Hoechst 33342 (nuclei stain: blue) (*N*=4). (C) RNA-seq analysis of CTL and *Nelfb*^−/− cells cultured in adipocyte differentiation medium for 14 days. Volcano plot shows 762 genes increased (red) and 358 genes decreased (blue) on a log$_2$ scale upon Nelfb depletion. *N*=3. (D) Gene Ontology (GO) terms of the 358 decreased and 762 increased genes using Enrichr. (E) Heat map of the 43 genes (≥2-fold change) found in the top five GO terms of decreased genes in D. (F) CUT&RUN on CTL and *Nelfb*^−/− cells cultured in adipocyte differentiation medium for 14 days using an anti-Nelfb and anti-IgG antibody. qPCR analysis was conducted on transcription start site (TSS) regions of *Pparg*, *Cebpa*, *Cebpb*, *Stat3* and *Krox20*. Signal was calculated as a percent of total input DNA (*N*=3). (G,H) CUT&RUN using an anti-RNA Polymerase II (RNA Pol II) antibody and anti-IgG antibody (*N*=3). qPCR analysis was performed on the TSS (G) or gene body (H) regions of *Pparg*, *Cebpa*, *Stat3* and *Krox20*. (I,J) CUT&RUN using an anti-RNA Polymerase II Ser2 phosphorylation (Pol II Ser2 Phospho) antibody and anti-IgG antibody (*N*=3). qPCR analysis was performed on the TSS (I) or gene body (J) regions of *Pparg*, *Cebpa*, *Stat3* and *Krox20*. (K) CUT&RUN using active chromatin mark anti-H3K4me3 and anti-IgG antibody. qPCR analysis on TSS regions of *Pparg*, *Cebpa*, *Stat3* and *Krox20*. Signal was calculated as a percent of total input DNA (*N*=3). Data are mean±s.e.m. *$P<0.05$, **$P<0.01$, ***$P<0.001$, ****$P<0.0001$ (one-way ANOVA followed by Tukey's multiple comparison tests). ns, not significant. Scale bars: 150 μm.

transduction groups: CTL cells transduced with the pBABE retrovirus (CTL pBABE), Nelfb$^{-/-}$ cells transduced with the pBABE retrovirus (Nelfb$^{-/-}$ pBABE), and Nelfb$^{-/-}$ cells transduced with pBABE expressing *Pparg* (Nelfb$^{-/-}$ Pparg). Following transduction, all groups underwent adipocyte differentiation induction for 14 days. Oil Red O staining showed prominent lipid-filled depots in the control cells (CTL pBABE), while the Nelfb-deleted cells lacked any staining (Nelfb$^{-/-}$ pBABE) (Fig. 3A,B). In contrast, Nelfb$^{-/-}$ cells expressing *Pparg* (Nelfb$^{-/-}$ Pparg) displayed robust Oil Red O staining, indicating that *Pparg* retroviral transduction successfully induced Nelfb$^{-/-}$ cells to differentiate into lipid-filled adipocytes (Fig. 3A,B). Plin1 staining revealed a marked increase in Plin1$^{+}$ adipocytes in the Nelfb$^{-/-}$ Pparg groups relative to the Nelfb$^{-/-}$ pBABE group (Fig. 3C-E). Interestingly, *Pparg* expression in Nelfb-deleted cells showed Plin1 levels comparable to control adipocytes (Fig. 3C-E). Staining for Pparg showed restored levels of the transcription factor in Nelfb-depleted cells expressing exogenous *Pparg*, which was comparable to levels observed in the control adipocytes (Fig. 3F-H). Gene expression analyses across the three groups, demonstrated that most of the adipogenic genes such as *Pparg*, *Cebpa*, *Plin1*, *Krox20*, *Adipoq*, *Acot4*, *Elovl6*, *Adig*, *Plaat3* and *Fabp4* were restored to levels comparable to the CTL adipocytes and significantly higher than the Nelfb$^{-/-}$ pBABE cells (Fig. 3I).This indicates that stable expression of exogenous *Pparg* not only enabled Nelfb-depleted cells to differentiate into adipocytes but also restored the expression of crucial adipogenic genes, effectively compensating for the loss of Nelfb.

### Nelfa and Nelfe are not necessary for adipocyte differentiation

To determine if control of adipocyte differentiation is applicable to all NELF complex members or is Nelfb specific, Nelfa and Nelfe were knocked down using siRNAs. Interestingly, loss of Nelfa resulted in the slight upregulation of differentiation genes such as *Acot4*, *Plaat3*, *Elovl6*, *Pparg*, *Cebpa*, *Plin1* and *Fabp4*, but downregulation of *Krox20* (Fig. S5A). Knockdown of Nelfe led to slight increases in the expression of *Acot4*, *Pparg*, *Cebpa* and *Fabp4*, but downregulation of *Adig* (Fig. S5B). Depletion of Nelfa did not alter the expression of Nelfe nor did knockdown of Nelfe impact Nelfa levels (Fig. S5A,B). Taken together, these results suggest that Nelfe and Nelfa are not necessary to promote adipocyte differentiation. Furthermore, there is a Nelfb-specific control of adipogenic differentiation. To investigate whether other NELF complex members bind to the same adipogenic genes as Nelfb, CUT&RUN using an antibody against Nelfe was performed in control and Nelfb-deleted cells. There was no significant binding of Nelfe to the TSS regions of *Pparg*, *Krox20* or *Cebpa* (Fig. S5C). However, there was binding of Nelfe to Stat3 in both control and Nelfb-depleted cells. These data suggest that Nelfb specifically binds to the adipogenic transcription factors, as other NELF complex members such as Nelfe do not bind to most of them.

### Short-term Rosi treatment restored dWAT formation and prolongs life in Nelfb$^{-/-}$ mice

Rosi is a selective Pparg agonist which plays a crucial role in adipogenesis and regulating insulin sensitivity in adipocytes (Gurnell, 2005). Upon Rosi binding to Pparg, an activated Pparg triggers the transcription of genes necessary for the differentiation of preadipocytes into mature adipocytes, promoting fat cell development and improving insulin responsiveness. As Nelfb$^{-/-}$ mice exhibit impaired adipogenesis and reduced dermal fat due to a decrease in *Pparg* expression, we hypothesized that activation of the remaining levels of Pparg with Rosi could restore adipocyte differentiation in these mice. To test this hypothesis, we administered Rosi (dissolved in DMSO and treated at

10 µg/g) or DMSO alone via intraperitoneal (IP) injection daily to pregnant female mice [approximately embryonic day (E)14] until birth, followed by topical application of Rosi or DMSO to the dorsal skin of the pups from P0 to P5 (Fig. 5A). On P6, the pups were euthanized for analysis (Fig. 5A). At P6, four groups of mice were analyzed: control mice treated with DMSO [+DMSO (CTL)], Nelfb$^{-/-}$ mice treated with DMSO [+DMSO (Nelfb$^{-/-}$)], control mice treated with Rosi [+Rosi (CTL)], and Nelfb$^{-/-}$ mice treated with Rosi [+Rosi (Nelfb$^{-/-}$)]. Oil Red O staining of dorsal skin showed multiple Oil Red O-positive depots in the dermis of +DMSO (CTL) mice or +Rosi (CTL), whereas +DMSO (Nelfb$^{-/-}$) mice displayed reduced staining (Fig. 4A,B). Incredibly, +Rosi (Nelfb$^{-/-}$) mice showed Oil Red O staining comparable to control mice (+/− Rosi) (Fig. 4A,B). These results suggest that short-term Rosi treatment facilitated the formation of lipid-filled adipocytes in the dermis of Nelfb$^{-/-}$ mice (Fig. 4A,B). Immunofluorescence, for Plin1 staining, revealed strong expression in the dermis of +DMSO (CTL) mice or +Rosi (CTL), whereas the protein was diminished in +DMSO (Nelfb$^{-/-}$) mice (Fig. 4C,D). In the presence of Rosi, Nelfb$^{-/-}$ mice were able to robustly express the mature adipocyte marker Plin1 (Fig. 4C,D). The +Rosi (Nelfb$^{-/-}$) mice were also larger and weighed significantly more than the +DMSO (Nelfb$^{-/-}$) mice (Fig. 4E,F). Gene expression analysis showed that many of the adipogenic genes such as *Acot4*, *Adig*, *Plaat3*, *Plin1*, *Adipoq* and *Fabp4* were significantly upregulated in the +Rosi (Nelfb$^{-/-}$) mice as compared to +DMSO (Nelfb$^{-/-}$) mice and reaching levels similar to those of +DMSO (CTL) mice (Fig. 4G). To evaluate the impact of short-term Rosi treatment on the lifespan of Nelfb-deleted mice, a survival analysis was performed (Fig. 5A,B). After IP injections of Rosi into pregnant female mice followed by topical application to pups from P0 to P5, we monitored their overall development and survival. Notably, the Rosi-treated Nelfb$^{-/-}$ mice survived for up to 15 days, while the untreated Nelfb$^{-/-}$ mice all died by P9 (Fig. 5B). Interestingly, Oil Red O staining of the dorsal skin of Rosi-treated Nelfb$^{-/-}$ mice that survived to days 14 and 15 showed that these mice had lost their dWAT (Fig. 5C,D). This suggests that continued treatment with the Pparg agonist Rosi is essential for maintenance of mature adipocytes in Nelfb-deleted mice (Fig. 5C,D).

### DISCUSSION

Our findings demonstrated that Nelfb is essential for dWAT development and adipocyte differentiation. Loss of Nelfb in Pdgfra$^{+}$ preadipocyte lineages led to failure of dWAT formation characterized by the absence of dermal adipocytes, which resulted in thinner skin and perinatal lethality. The absence of mature adipocytes in Nelfb$^{-/-}$ mice was not due to a depletion of progenitor cells, as Sca1$^{+}$ Pdgfra$^{+}$ preadipocyte populations remained unchanged. Rather, Nelfb deletion prevented the differentiation of these precursors into mature lipid-filled adipocytes. This was further supported by *in vitro* differentiation assays, which showed that Nelfb-depleted cells failed to undergo adipogenic differentiation, as evidenced by their inability to accumulate lipid droplets and the reduced expression of adipogenic markers. Our RNA-seq data revealed significant downregulation of key adipogenic transcription factors (*Pparg* and *Cebpa*) suggesting that Nelfb is necessary for the proper transcriptional activation of genes involved in adipocyte differentiation. Pathway enrichment analysis showed that genes involved in fatty acid metabolism, Pparg signaling and lipid biosynthesis were significantly repressed, whereas genes associated with Wnt signaling and the IL6 signaling pathway were upregulated. Wnt pathway genes are well-described inhibitors of adipogenesis and are highly expressed in progenitor cells to prevent differentiation (Christodoulides et al., 2009; Yang Loureiro et al.,

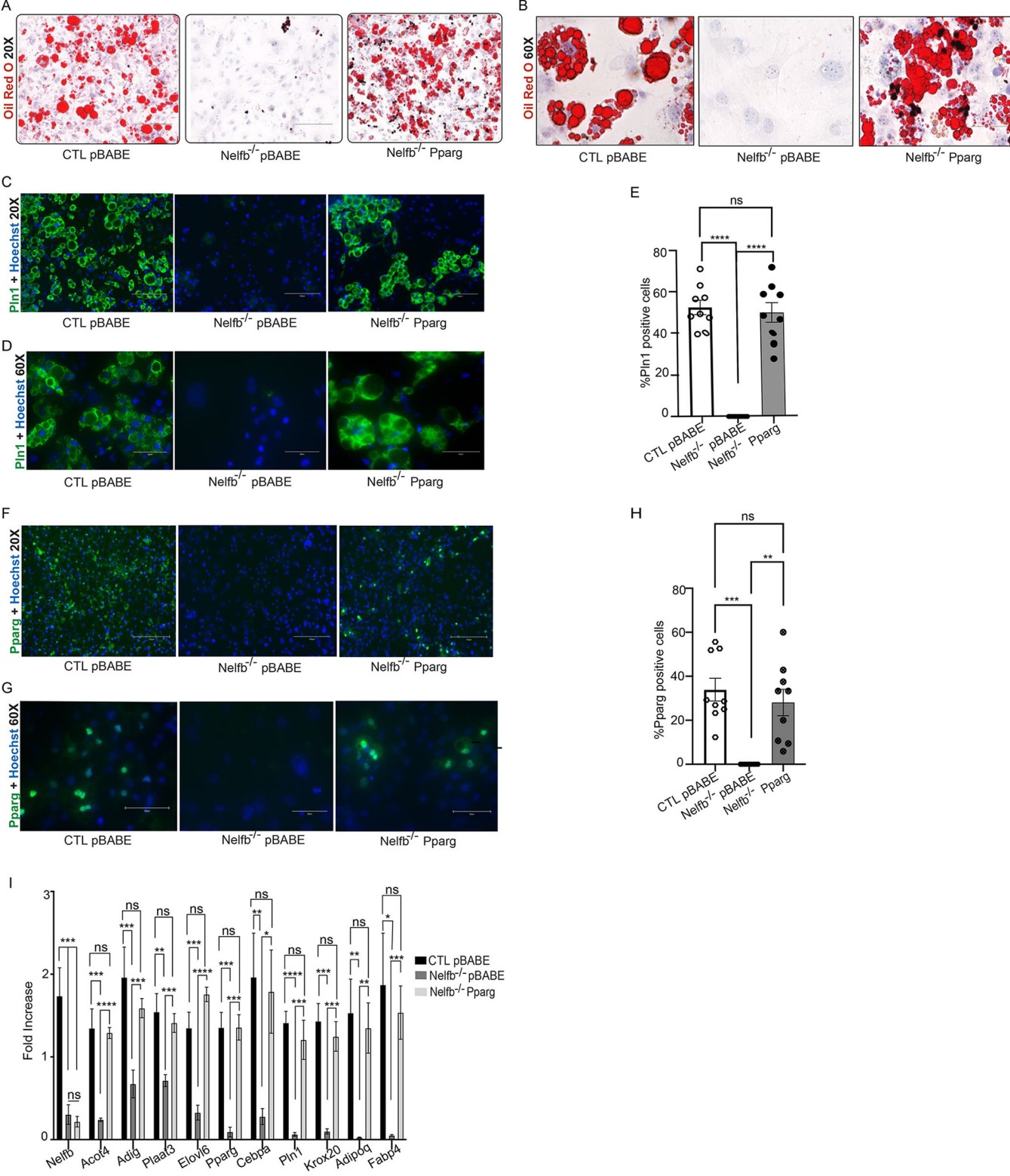

**Fig. 3. Exogenous expression of *Pparg* restores adipocyte differentiation in Nelfb deleted cells.** (A,B) Oil Red O staining of control cells transduced with pBABE retrovirus (CTL pBABE), *Nelfb*−/− cells transduced with pBABE retrovirus (*Nelfb*−/− pBABE) and *Nelfb*−/− cells transduced with Pparg retrovirus (*Nelfb*−/− Pparg). Cells were cultured in adipocyte differentiation medium for 14 days. Images were taken at 20× (A) and 60× (B) magnification (*N*=3). (C,D) Staining of CTL pBABE, *Nelfb*−/− pBABE and *Nelfb*−/− Pparg cells with antibodies against perilipin 1 (Plin1; mature adipocyte marker: green) and Hoechst 33342 (nuclei: blue) at 20× (C) and 60× (D) magnification (*N*=3). (E) Percent of Plin1+ cells quantified from D. (F,G) Staining of CTL pBABE, *Nelfb*−/− pBABE and *Nelfb*−/− Pparg with antibodies against Pparg (green) and Hoechst 33342 (blue) at 20× (F) and 60× (G) magnification (*N*=3). (H) Percent of Pparg+ cells quantified from G. (I) RT-qPCR of adipogenic genes on CTL pBABE, *Nelfb*−/− pBABE and *Nelfb*−/− Pparg cells (*N*=4). Results were normalized to housekeeping gene, *Gapdh*. Data are mean±s.e.m. *$P<0.05$, **$P<0.01$, ***$P<0.001$, ****$P<0.0001$ (RM one-way ANOVA followed by Tukey's multiple comparison tests for E and H; multiple unpaired t-tests for I). ns, not significant. Scale bars: 150 μm (A,C,F); 50 μm (B,D,G).

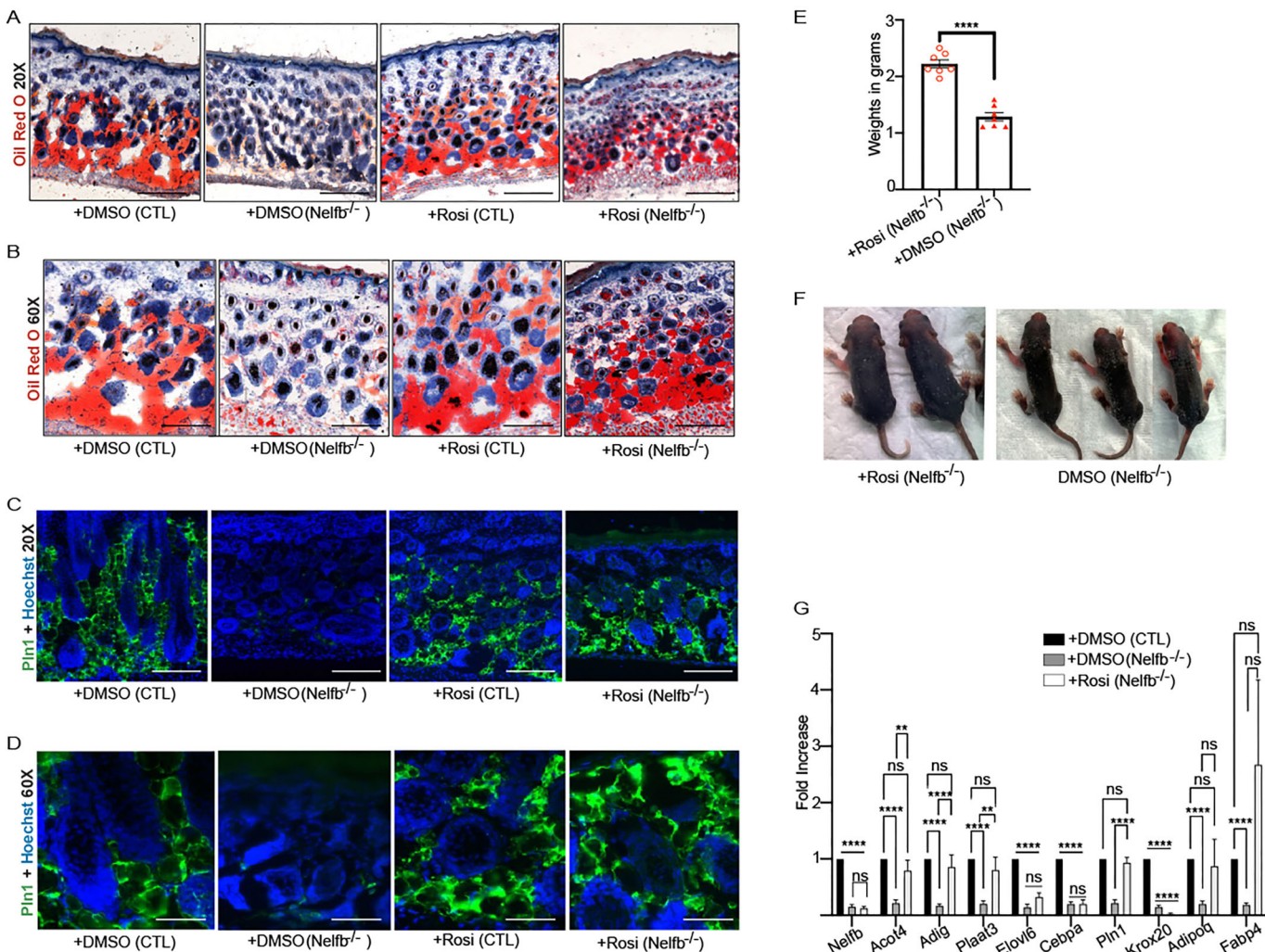

**Fig. 4. Rosi treatment restores dWAT formation in Nelfb⁻/⁻ mice.** (A,B) Oil Red O staining of dorsal skin from control (CTL: *Nelfb^fl/wt Pdgfra-Cre*⁺ or *Nelfb^wt/wt Pdgfra-Cre*⁺) mice treated with DMSO [+DMSO(CTL)] and Rosi [+Rosi(CTL)], or *Nelfb*⁻/⁻ (Nelfb^fl/fl Pdgfra-Cre⁺) mice treated with DMSO [+DMSO(*Nelfb*⁻/⁻)] and Rosi [+Rosi(*Nelfb*⁻/⁻)]. Animals were harvested at P6. Images were taken at 20× (A) and 60× (B) magnification. (*N*=7) (C,D) Staining of dorsal skin with antibodies against Plin1 (green) and Hoechst 33342 (nuclei: blue) at 20× (C) and 60× (D) magnification (*N*=7). (E) Body weights of *Nelfb*⁻/⁻ mice treated with DMSO or Rosi were measured at P6 (*N*=7). Each dot represents weight of a single animal. (F) Representative images of P6 *Nelfb*⁻/⁻ mice treated with DMSO or Rosi. (G) RT-qPCR of adipogenic genes on dermis harvested from +DMSO(CTL), +DMSO(*Nelfb*⁻/⁻) and +Rosi(*Nelfb*⁻/⁻) mice at P6 (*N*=4). Results were normalized to housekeeping gene, *Gapdh*. **$P<0.01$, ****$P<0.0001$ (unpaired *t*-tests for E; multiple unpaired *t*-tests for G). ns, not significant. Data are mean±s.e.m. Scale bars: 150 μm (A,C); 50 μm (B,D).

2023). The upregulation of Wnt-related genes such as *Wnt10b*, *Sox9* and *Dlk1* in Nelfb⁻/⁻ cells suggested that these cells are trapped in a progenitor state and are unable to differentiate. In support of Nelfb-deleted cells being locked in the progenitor state, *IL6*, a lipolytic cytokine known to be expressed in preadipocytes (Radvanyi and Roszer, 2024; Harkins et al., 2004), was also upregulated in Nelfb⁻/⁻ cells. The increased expression of *IL6* in the absence of Nelfb suggested a potential shift toward a catabolic adipose tissue phenotype, further contributing to the loss of fat phenotype observed in Nelfb-deficient mice. To determine whether Nelfb directly regulated and impacted RNA Pol II stability on adipogenic regulators, we used CUT&RUN. Notably, Nelfb and RNA Pol II bound at both the TSS and gene bodies of transcription factors such as *Pparg*, *Cebpa*, *Stat3* and *Krox20* in adipocytes. We observed a loss of Nelfb and RNA Pol II binding at the TSS and gene body of *Pparg*, *Cebpa*, *Stat3* and *Krox20* in Nelfb⁻/⁻ cells. In addition, loss of Nelfb also resulted in decreased H3K27Ac and H3K4me3 marks at the TSS of the adipogenic genes, concomitant with an increase in repressive

histone modifications such as H3K9me3 and H3K27me3. Nelfb loss potentially causes compacted chromatin due to the loss of RNA Pol II binding and thus loss of transcription. The process of RNA Pol II-mediated transcription is essential for chromatin decompaction and loss of it leads to chromatin closure (Gu et al., 2018; Jubb et al., 2017). Previous studies using small molecule inhibitors of RNA Pol II have shown that transcription directly promotes open chromatin through displacement of cohesin binding (Heinz et al., 2018; Busslinger et al., 2017). However, a study using acute degradation of RNA Pol II using the auxin inducible degron system showed that loss of transcription indirectly caused chromatin closure (Jiang et al., 2020). It is currently unclear whether the chromatin closure seen in the adipogenic genes due to Nelfb loss is a direct or indirect consequence of reduced transcription and would be an area of interest for future investigation. Interestingly, Nelfb and RNA Pol II were already bound to the TSS of *Pparg*, *Cebpa*, *Stat3* and *Krox20* in adipocyte progenitor cells. However, there were low levels of RNA Pol II detected in the gene bodies, suggesting that RNA Pol II was paused at the TSS in

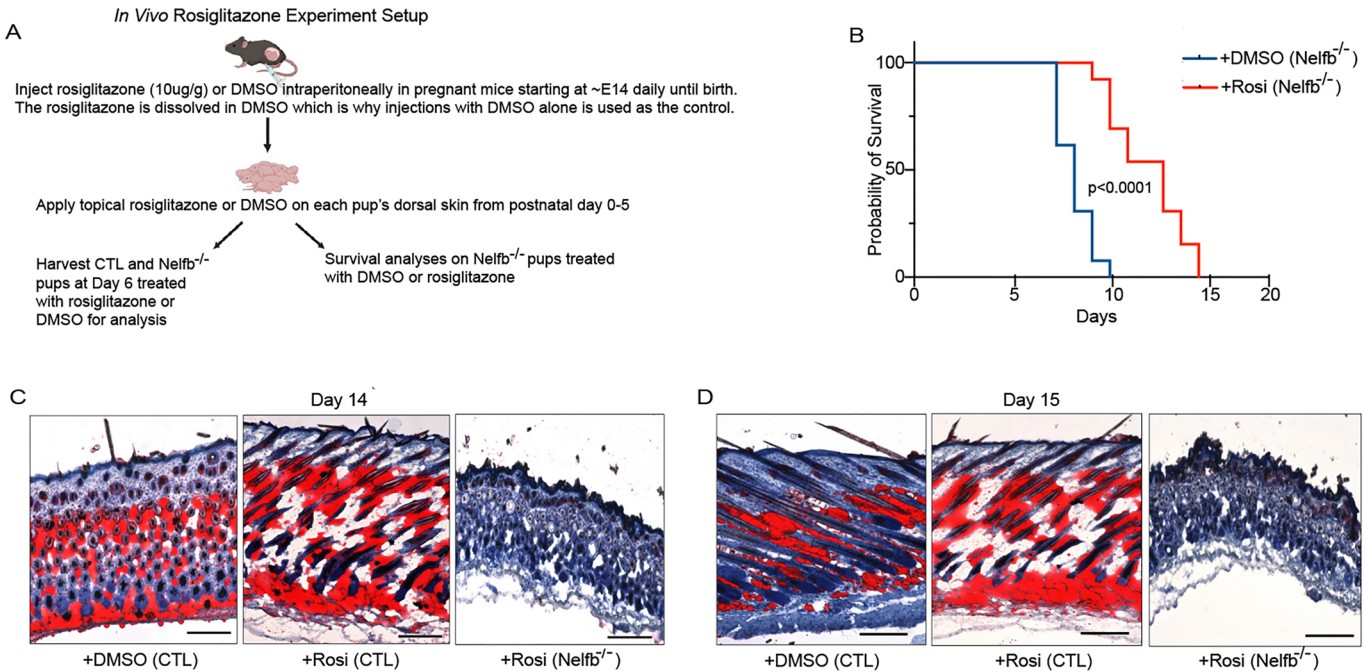

**Fig. 5. Rosi treatment extends the life of Nelfb deleted mice.** (A) Schematic of Rosi administration in pregnant mice followed by topical skin application of Rosi on CTL and *Nelfb*$^{-/-}$ pups from P0-P5. Mice were harvested at P6 for further analyses or used for survival analysis. (B) Survival curve analysis of *Nelfb*$^{-/-}$ mice treated with DMSO [+DMSO(*Nelfb*$^{-/-}$)] or Rosi [+Rosi(*Nelfb*$^{-/-}$)] (*N*=12). *P*<0.0001 (Gehan-Breslow-Wilcoxon test). Mice were treated with DMSO or Rosi as illustrated in A. (C,D) Oil Red O staining of dorsal skin from Rosi-treated *Nelfb*$^{-/-}$ mice that survived until P14 (C) or P15 (D). Scale bars: 300 µm.

progenitor cells that were subsequently released into productive elongation during adipocyte differentiation. In contrast to the importance of Nelfb for RNA Pol II stabilization on adipogenic differentiation genes, we found that Nelfb is not required for the transcription of preadipocyte identity genes. RNA Pol II was bound at the TSS of preadipocyte genes such as *Dlk1*, *Pbx1*, *Sox9* and *Meis1* in both control and Nelfb-deleted progenitor cells. This indicates that these genes remain actively transcribed despite the absence of Nelfb, demonstrating that Nelfb is not required for RNA Pol II stabilization of preadipocyte-specific transcription. Taken together, these findings highlight the role of Nelfb in regulating the transcription of mature adipocyte genes, and the fact that it is dispensable for preadipocyte gene expression. It is still unclear why Nelfb deletion impacts only a subset of genes with such cell type or cell state specificity despite it being an active area of research (Core and Adelman, 2019; Robinson et al., 2021). This type of specificity has recently been shown in muscle stem cells, where Nelfb deletion resulted in the loss of expression of genes essential for promoting expansion of progenitor cells during regeneration (Robinson et al., 2021). This resulted in premature differentiation of progenitor cells, which caused inefficient regeneration after muscle damage. Similarly, deletion of Nelfb in CD8 T cells led to loss of expression of TCF1, which inhibited proliferation and resulted in enhanced differentiation (Wu et al., 2022). It is possible that the most sensitive Nelfb-regulated genes may be ones that require robust pausing and elongation, such as developmental or proliferation genes necessary for cell state transitions. It is also worth noting that there is a Nelfb-specific requirement for promoting adipocyte differentiation through RNA Pol II stabilization. Nelfa or Nelfe knockdown using siRNAs did not block differentiation gene expression but instead slightly increased it. In addition, Nelfe did not bind a majority of the genes that Nelfb binds and regulates. Other reports in which deletion of Nelfb has been shown to have a functional role in T cells, muscle stem cells and cardiomyocytes have

all attributed the phenotype to the entire NELF complex (Wu et al., 2022; Pan et al., 2014; Robinson et al., 2021). However, the other subunits were not targeted in these studies and may be worth exploring to determine if the phenotypes were Nelfb specific or can be attributed to the broader NELF complex. Given that Nelfb regulates the expression of *Pparg* (Rosen et al., 1999), a master regulator of adipogenesis, we wanted to determine if restoration of *Pparg* expression could drive differentiation in Nelfb$^{-/-}$ cells. Importantly, exogenous, retroviral *Pparg* expression was sufficient to restore adipocyte differentiation in Nelfb$^{-/-}$ cells. This suggests that Nelfb control of differentiation primarily acts through Pparg. Furthermore, our *in vivo* data showed that pharmacological activation of the remaining levels of Pparg via Rosi treatment partially restored dWAT formation and extended the survival of Nelfb$^{-/-}$ mice. This highlights a potential therapeutic strategy for overcoming impaired adipogenesis in dWAT. Similar approaches have been used to treat the atrophy of dWAT due to anti-epidermal growth factor receptor (EGFR) therapy. EGFR is a therapeutic target for a variety of cancers, however anti-EGFR therapy often leads to cutaneous toxicity characterized by papulopustular rashes with atrophy of dWAT. A recent study using a mouse model of anti-EGFR therapy showed that Rosi treatment led to the expansion of dWAT, decreasing the severity of rashes (Chen et al., 2022). Our study also demonstrated that Nelfb loss led to diminished brown and white adipose tissue, suggesting the importance of transcription pause/elongation regulation in these tissues. Thus, Nelfb deletion from Pdgfra$^+$ cells offers a unique model to investigate Nelfb function in different adipose depots, which may reveal tissue-specific transcriptional mechanisms regulating adipogenesis.

In summary, our study identifies Nelfb as a crucial transcriptional regulator of dWAT formation. Through its role in RNA Pol II stabilization and maintenance of open chromatin, Nelfb ensures the proper expression of adipogenic transcription factors necessary for adipocyte differentiation. Investigating whether Nelfb function can be

modulated to treat disorders of adipose tissue development may offer new therapeutic opportunities for metabolic diseases and regenerative medicine.

## MATERIALS AND METHODS
### Mice
Nelfb floxed (The Jackson Laboratory, #033115) mice were crossed with Pdgfra-Cre (The Jackson Laboratory, #013148) transgenic mice to generate control (Nelfb$^{fl/wt}$ Pdgfra-Cre$^{+/-}$ or Nelfb$^{wt/wt}$ Pdgfra-Cre$^{+/-}$) or Nelfb deleted (Nelfb$^{fl/fl}$ Pdgfra-Cre$^{+/-}$) mice. The mice were maintained on a C57BL/6J genetic background. Genotyping was performed by PCR using genomic DNA extracted from tail biopsies. Specific primer sets were designed to detect the presence of the loxP-flanked Nelfb allele (450 bp), the wild-type allele (400 bp), and the Cre (370 bp) transgene. PCR reactions were carried out using Q5 Hot Start 2× Master Mix (New England Biolabs, #M0271S) under the following conditions: Initial denaturation was performed at 95°C for 5 min, followed by 30 cycles of denaturation at 95°C for 30 s, annealing at 58°C for 30 s, and extension at 72°C for 45 s. A final extension step was conducted at 68°C for 5 min, and samples were subsequently held at 4°C indefinitely. Each 20 µl reaction consisted of 10 µl of PCR mix, 5 µl of dH$_2$O, 1 µl of each primer, and 4 µl of DNA template. PCR products were analyzed using 1% agarose gel electrophoresis. Control and Nelfb$^{-/-}$ mice were harvested at Days 6 and 9 for downstream experiments.

### Primers and siRNA sequences
All primers and siRNA sequences can be found in Table S2.

### Neonatal fibroblasts and adipocyte differentiation
Dermal fibroblasts were isolated from the dorsal skin of P0 neonatal mice. Dorsal skin from control (Nelfb$^{fl/wt}$ Pdgfra-Cre$^{+/-}$ or Nelfb$^{wt/wt}$ Pdgfra-Cre$^{+/-}$) or Nelfb-deleted (Nelfb$^{fl/fl}$ Pdgfra-Cre$^{+/-}$) mice was placed into Eppendorf tubes containing 2 ml of 3 mg/ml dispase dissolved in EpiLife media (Life Technologies, #MEPI500CA), supplemented with EDGS (Life Technologies, #S0015) and penicillin/streptomycin (pen/strep). The tubes were incubated overnight at 4°C on a tube rocker. Following incubation, the dorsal skin was removed from the tubes, and the epidermis was peeled away while keeping the dermis side down. The dermis was washed in PBS, then minced using a scalpel in a 10 cm dish containing 1-2 ml of filtered HBSS buffer supplemented with 20 mg/ml bovine serum albumin (BSA) (Sigma-Aldrich, #A7906) and 1× pen/strep, kept on ice. The minced dermal tissue was transferred into a 15 ml conical tube, and 5× Collagenase D solution (2.5 mg/ml, Roche, #11088882001) and 1 mg/ml DNase I (1:30, Sigma-Aldrich, #D5025) were added. The 15 ml conical tubes were warmed in a 37°C water bath for 5 min, then incubated at 37°C with rotation for 1-2 h. After incubation, the tubes were vortexed and placed on ice for 10 min. The suspension was then passed sequentially through 100 µm and 40 µm filters to remove hair, fat and cell clumps. The flow-through was centrifuged at 1400 rpm (368 *g*) for 10 min at 4°C, and the supernatant was discarded. The pellet was resuspended in complete DMEM media (Gibco, #10-013-CV) supplemented with 10% fetal bovine serum (FBS; Gibco, #A5256701) and pen/strep. A 10 µl aliquot of the cell suspension was taken for cell counting, and the remaining cells were plated in 12 ml of DMEM media in a 10 cm dish. The media was replenished 24 h later. Once the cells reached 70% confluency, they were trypsinized, counted and plated into 12-well plates at a density of 700,000-800,000 cells per well in DMEM media (growth media). Adipocyte differentiation was induced by adding differentiation media (Stemcell Technologies, #05507) supplemented with adipocyte supplement and pen/strep. Cells were differentiated for 14 days, after which downstream analyses were performed.

### *In vivo* Rosi treatment
Pregnant mice received daily IP injections of either 10 µg/g Rosi (Sigma-Aldrich, #R2408) or DMSO (Sigma-Aldrich, #472301), starting at ~E14 and continuing until birth. Rosi was dissolved in DMSO, while the control group received DMSO alone. Following birth, Rosi or DMSO was applied topically to the dorsal skin of each pup from P0 through to P5. On P6, control and Nelfb$^{-/-}$ mice treated with either Rosi or DMSO were used for survival analysis or harvested for downstream assays.

### Dermal isolation
On P6 or P9, dorsal skin from control and Nelfb$^{-/-}$ mice was isolated and washed with PBS and incubated overnight at 4°C, dermis side down, in a dispase solution (Gibco, #17105-041) (3 mg/ml in HBSS buffer). Following incubation, the epidermis was carefully separated from the dermis and discarded. The dermis was then washed in PBS, finely minced and harvested for RNA extraction.

### RNA isolation and RT-qPCR
Total RNA was extracted from cells using the GeneJET RNA Purification Kit (Thermo Fisher Scientific, #K0732) and RNA extracted from tissue using the RNeasy plus mini kit (Qiagen, #74136). RNA concentration was measured using a Nanodrop spectrophotometer. Then 1 µg of total RNA was reverse-transcribed into cDNA using the Maxima cDNA Synthesis Kit (Thermo Fisher Scientific, #K1642). Quantitative PCR was conducted using the BioRad LFX96 real-time PCR system. *Gapdh* was used as an internal control for normalization for both cells and dorsal skin tissue.

### Immunofluorescence
For immunofluorescence, 10 µm dorsal skin tissue was fixed in 4% paraformaldehyde (PFA) for 11 min, followed by a 30-min blocking step in PBS containing 2.5% normal goat serum, 0.3% Triton X-100 and 2% BSA. Primary antibodies used were perilipin 1 (Pln-1, Cell Signaling Technology, #D1D8, 1:300) and peroxisome proliferator-activated receptor gamma (PPARγ, Cell Signaling Technology, #C26H12, 1:300). Secondary antibodies used were Alexa 555-conjugated goat anti-rabbit IgG (Life Technologies, #A21428, 1:500) and Alexa 488-conjugated donkey anti-rabbit IgG (Life Technologies, #A21206, 1:500). Nuclei were counterstained with Hoechst 33342 (Thermo Fisher Scientific, #H3570) at 1:1000. Samples were mounted with Fluoromount-G (Southern Biotech, #0100-01) and fluorescence images were acquired using the EVOS M5000 Imaging System (Thermo Fisher Scientific). For immunofluorescence staining of cultured cells, control and Nelfb$^{-/-}$ cells grown in adipocyte differentiation media in 12-well plates were fixed in 4% PFA for 10 min. After three washes with PBS containing 0.1% Tween (3 min per wash), cells were permeabilized in 0.2% Triton X-100 in PBS for 10-20 min. Blocking was performed in PBS with 5% goat serum for 1 h. Cells were then incubated overnight at 4°C with primary antibodies against Plin1 (Cell Signaling Technology, #D1D8, 1:100), Ki67 (Abcam, #AB15580, 1:100) and PPARγ (Cell Signaling Technology, #C26H12, 1:100). Following three washes with PBS+0.1% Tween, the secondary antibody Alexa 488-conjugated donkey anti-rabbit IgG was applied at 1:500 for 45 min at room temperature. Cells were then washed three times with PBS, counterstained with DAPI for 2 min and rinsed briefly with PBS followed by MilliQ water to remove residual salts. Fluoromount-G was used for mounting and circular coverslips were placed over the wells. Imaging was performed using the EVOS M5000 Imaging System.

### Oil Red O staining
Dorsal skin sections (10 µm) were fixed in 10% formalin for 15 min, followed by a 5-min incubation in distilled water. Slides were then treated with 60% isopropanol for 15 min before staining with Oil Red O for 15 min. The stock solution was prepared by dissolving 0.5 g of Oil Red O (Sigma-Aldrich, #O0625) in 100 ml of 100% isopropanol (Sigma-Aldrich, #I9030-4L). For working solution preparation, 15 ml of the stock was diluted with 10 ml of distilled water, incubated at room temperature for 10 min and filtered into a Coplin jar. After staining, slides were dipped ten times in 60% isopropanol, followed by two 3-min washes in distilled water. Hematoxylin (VectorLabs, #H3401) staining was performed for 3-5 min, followed by two 5-min washes in distilled water. Finally, slides were mounted with Fluoromount-G and visualized under bright field using the EVOS M5000 Imaging System.

### Hematoxylin and Eosin staining
Dorsal skin sections (20 µm) were fixed in 10% formalin for 12 min at room temperature, followed by permeabilization in 0.25% Triton X-100 in PBS for 5 min. Slides were then washed with distilled water and incubated in hematoxylin solution for 1 min at room temperature. Excess hematoxylin was removed by rinsing in distilled water. Slides were dipped ten times in 0.3% acid alcohol, followed by incubation in 0.2% ammonia water for 2 min. After

thorough washing in distilled water to remove ammonia, slides were dipped ten times in 95% ethanol, then incubated in Eosin Y stain (Sigma-Aldrich, #HT110116) for 20 s. This was followed by another ten dips in 95% ethanol and two sequential incubations in 100% ethanol for 5 min each. Finally, slides were cleared in Xylene (Sigma-Aldrich, #214736) (2× for 3 min each), mounted with Permount (Sigma-Aldrich, #SP15-500) and visualized under bright field using the EVOS M5000 Imaging System.

### CUT&RUN

Control and Nelfb$^{-/-}$ cells were cultured in adipocyte differentiation medium for 14 days: ∼120,000-140,000 cells/reaction were trypsinized, counted and pelleted. The CUT&RUN kit used for these experiments was purchased from Cell Signaling Technology (#91931). Pellets were fixed in 16% PFA for 2 min, and cross-linking was stopped using 10× glycine for 5 min at room temperature. Cells were washed with 1× Wash Buffer after centrifugation (3000× $g$, 3 min, 4°C). Each reaction received 100 µl 1× Wash Buffer and resuspended pellets. Concanavalin buffer (10 µl per 10 µl magnetic beads/reaction) was added, incubated for 5 min, and removed using a magnetic rack. Cell suspensions were incubated with antibodies in 100 µl Antibody Binding Buffer overnight at 4°C on a rotator. Quantities used were: Nelfb: 5 µl/reaction (Cell Signaling Technology, #14894S); RNA Pol II: 3 µl/reaction (Active Motif, #39097); RNA Pol II Ser2 Phosphorylation: 2 µl/reaction (Cell Signaling Technology, #13499S); Nelfe: 5 µl/reaction (Proteintech, #10705-1-AP); IgG: 5 µl/reaction (Cell Signaling Technology, #66362); H3K4me3: 5 µl/reaction (Cell Signaling Technology, #9751S); H3K27me3: 2 µl/reaction (Cell Signaling Technology, #9733S); H3K9me3: 2 µl/reaction (Cell Signaling Technology, #13969S); H3K27Ac: 1 µl/reaction (Cell Signaling Technology, #8173S). After washing with Digitonin Buffer, 50 µl pAG-MNase pre-mix was added to each reaction and incubated at 4°C for 1 h. MNase activation was triggered with 3 µl cold CaCl$_2$, incubated at 4°C for 30 min, and terminated by adding 150 µl Stop Buffer and 5 µl Spike-In DNA (this was added for sample normalization) per sample followed by 10 min incubation at 37°C. DNA fragments were collected by centrifugation (16,000 $g$ at 4°C for 2 min) and the supernatant transferred to new tubes. Samples were incubated at 65°C for 2 h with SDS and Proteinase K to reverse crosslinks. DNA was purified using DNA purification buffers and spin columns kit (Cell Signaling Technology, #14209). Input samples were mixed with Extraction Buffer (Proteinase K+RNAse A) and incubated at 55°C for 1 h. After cooling on ice for 5 min, chromatin was fragmented via sonication (three cycles: 7.5 min total, 0.5 s off, 1 s on). Lysates were clarified (18,500× $g$, 10 min, 4°C) and DNA was purified. Gel electrophoresis (2-4% agarose) confirmed sheared DNA fragments between 200-400 bp.

### RNA-seq and library preparation

Biological triplicates of control and Nelfb$^{-/-}$ cells cultured in adipocyte differentiation medium for 14 days were collected for total RNA extraction using the GeneJET RNA purification kit (Thermo Fisher Scientific, #K0732) and quantified via Nanodrop. RNA-seq was conducted on the Illumina NovaSeq S4 platform at the University of California, San Diego Institute of Genomic Medicine core facility. Libraries were prepared using Illumina RNA Seq kit (Illumina, #20040534), multiplexed, and sequenced to generate ∼25 million reads per sample. The RNA-seq data have been deposited in GEO with accession number GSE291112.

### Retroviral transduction of *Pparg*

Amphotropic Phoenix cells were transfected with 3 µg of each retroviral expression construct using Lipofectamine 2000 (Life Technologies, #11668019). The pBABE puro PPARγ2 plasmid, a gift from Bruce Spiegelman (Tontonoz et al., 1994; Addgene plasmid #8859), was used for *Pparg* overexpression, while the empty pBABE puro plasmid (Addgene plasmid #110349) served as a negative control. After 24 h, the culture medium was replaced, and viral supernatants were collected three times. For viral transduction, 110,000-125,000 P0 fibroblasts were seeded in six-well plates and infected with either pBABE control or pBABE-PPARG retrovirus. Viral supernatants were supplemented with 5 µg/ml Polybrene (Sigma-Aldrich, #H9268) per well, and plates were centrifuged at 1000 rpm (188 $g$) for 1 h. Transduction was repeated the following day. To select for successfully transduced cells, puromycin was added to the culture. Surviving

cells were then induced for adipocyte differentiation over 14 days, after which they were harvested for downstream analyses.

### Knockdown of Nelfa and Nelfe

Neonatal fibroblasts were harvested from control wild-type mice and plated in 12-well plates. Cells were cultured until they reached ∼60-70% confluency. For transfection, Lipofectamine™ RNAiMAX reagent (Thermo Fisher Scientific, #13778030) was diluted in Opti-MEM™ medium (Thermo Fisher Scientific, #31985070); 9 µl Lipofectamine was added to 150 µl Opti-MEM per well (Solution A; total volume 159 µl/well). Separately, Nelfa, Nelfe or non-targeting control siRNAs (Invitrogen) were each diluted in Opti-MEM (6 µl siRNA+150 µl Opti-MEM per well; Solution B; total volume 156 µl/well). Solutions A and B were combined at a 1:1 ratio, gently mixed, and incubated for 20 min at room temperature to allow complex formation. The siRNA–lipid complexes were then added dropwise to the appropriate wells containing DMEM supplemented with 10% FBS. Plates were incubated at 37°C for 24 h, after which cells were trypsinized and replated into 10 cm dishes. The medium was replaced with adipocyte differentiation medium 24 h later, and cells were differentiated for 7 days. On P7, cells were harvested for RNA extraction.

### Flow cytometry

Mouse skin was minced into small fragments and enzymatically digested at 37°C for 1 h and 15 min in a digestion buffer containing DMEM/high glucose medium, 10% FBS, 0.35 mg/ml Liberase TL (Roche, #5401020001), 3 mg/ml Collagenase D (Roche, #11088882001) and 0.1 mg/ml DNase I (Sigma-Aldrich, #D5025). The resulting cell suspension was filtered through a 70 µm strainer to obtain single cells. After centrifugation at 2000 rpm (751 $g$) for 10 min at 4°C, the pellet was resuspended in FACS buffer (PBS with 3% FBS and 5 mM EDTA) and blocked with FC blocking buffer (eBiosciences, #14-0161-82) for 15 min. Following centrifugation and removal of the supernatant, cells were stained with CD45-FITC (BioLegend, #103108), CD31-PE (BioLegend, #102507), Sca1-PE/Cyanine 7 (BioLegend, #108113) and Pdgfra-APC (eBiosciences, #17-1401-81) antibodies at a 1:100 dilution and incubated in the dark on ice for 30 min. Cells were then centrifuged at 2000 rpm (751 $g$) for 3 min at 4°C, followed by two PBS washes. Viability dye eFluor 506 was used to distinguish live from dead cells. Flow cytometry data were acquired using a BD LSRFORTESSA and analyzed with FlowJo 10.8.1 software.

### RNA-seq data processing

Reads were aligned to the mm10 mouse aligner index – whole genome – using DeSeq2 (Love et al., 2014) with default settings. Differential expression was assessed via ANOVA in the Partek Genomic Suite (Partek Incorporated). A read count threshold of ten reads per gene was used to distinguish expressed from unexpressed genes, and genes below this threshold were excluded from ANOVA analysis. Significantly upregulated or downregulated genes were identified based on FDR≤0.05 and ≥2-fold change. Enriched GO terms for differentially expressed genes were determined using Enrichr (Kuleshov et al., 2016). RNA-seq heatmaps using normalized counts were generated using R-Studio software.

### Analysis of transcription factor binding sites

The promoter regions (−500 bp to TSS) of *Pparg*, *Cebpa*, *Stat3* and *Krox20* were scanned for transcription factor binding motifs using TRANSFAC (Wingender et al., 2000). The only transcription factor motif that was found in all four genes was Sp1. The number of Sp1 binding sites were also counted for each gene.

### Statistical analysis

Significant changes ($P<0.05$) were determined by $t$-test (groups of 2) or ANOVA (groups >2). Gehan-Breslow-Wilcoxon test was used for survival analysis.

### Acknowledgements

RNA-seq was conducted at the IGM Genomics Center, University of California, San Diego, La Jolla, CA, USA.

**Competing interests**
The authors declare no competing or financial interests.

**Author contributions**
Conceptualization: G.L.S., S.M.; Data curation: S.M., J.G., U.B., Y.L., C.F.-M., Y.C.; Formal analysis: S.M.; Funding acquisition: G.L.S.; Investigation: S.M., Y.L., C.F.-M., Y.C.; Methodology: S.M., J.G., U.B., C.F.-M., Y.C.; Project administration: S.M.; Supervision: G.L.S.; Validation: S.M.; Writing – original draft: S.M.; Writing – review & editing: G.L.S., S.M., J.G., U.B., Y.L., C.F.-M.

**Funding**
This study was supported by grants from the National Institute of Arthritis and Musculoskeletal and Skin Diseases (R01AR081215 and R01AR066530) and the National Cancer Institute (R01CA269303) to G.L.S. Open Access funding provided by University of California, San Diego. Deposited in PMC for immediate release.

**Data and resource availability**
The RNA-seq dataset has been deposited in GEO under accession number: GSE291112. All other relevant data and details of resources can be found within the article and its supplementary information.

**Peer review history**
The peer review history is available online at https://journals.biologists.com/dev/lookup/doi/10.1242/dev.204976.reviewer-comments.pdf

**Special Issue**
This article is part of the Special Issue 'Lifelong Development: the Maintenance, Regeneration and Plasticity of Tissues', edited by Meritxell Huch and Mansi Srivastava. See related articles at https://journals.biologists.com/dev/issue/152/20.

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
