## [Peer Review File · Development (Cambridge, England)]

Nelfb promotes dermal white adipose tissue formation through RNA polymerase II-mediated adipogenic gene regulation

Samiksha Mahapatra, Julian Gomez, Uyanga Batzorig, Ye Liu, Celia Fernández-Méndez, Yifang Chen and George L. Sen
DOI: 10.1242/dev.204976

Editor: Haruhiko Koseki

Review timeline

Original submission:	27 May 2025
Editorial decision:	28 July 2025
First revision received:	28 August 2025
Accepted:	7 September 2025

Original submission

First decision letter

MS ID#: dev.204976

MS TITLE: Nelfb Promotes Dermal White Adipose Tissue Formation through RNA Polymerase II Mediated Adipogenic Gene Regulation

AUTHORS: George Sen; Samiksha Mahapatra; Julian Gomez; Uyanga Batzorig; Ye Liu; Celia Fernandez-Mendez; Yifang Chen

Dear Dr Sen,

I have now received all the referees' reports on the above manuscript, and have reached a decision. The referees' comments are appended below, or you can access them online: please go to:

As you will see, the referees express considerable interest in your work, but have some significant criticisms and recommend a substantial revision of your manuscript before we can consider publication. If you are able to revise the manuscript along the lines suggested, which may involve further experiments, I will be happy receive a revised version of the manuscript. Your revised paper will be re-reviewed by one or more of the original referees, and acceptance of your manuscript will depend on your addressing satisfactorily the reviewers' major concerns. Please also note that Development will normally permit only one round of major revision. If it would be helpful, you are welcome to contact us to discuss your revision in greater detail. Please send us a point-by-point response indicating your plans for addressing the referees' comments, and we will look over this and provide further guidance.

Please attend to all of the reviewers' comments and ensure that you clearly highlight all changes made in the revised manuscript. Please avoid using 'Tracked changes' in Word files as these are lost in PDF conversion. I should be grateful if you would also provide a point-by-point response detailing how you have dealt with the points raised by the reviewers in the 'Response to Reviewers' box. If you do not agree with any of their criticisms or suggestions please explain clearly why this is so.

Reviewer 1

The manuscript by Mahapatra et al. characterize the dermal loss of Nelfb and find that dWAT is substantially reduced, along with relevant adipogenic gene expression, and mice die early. Through elegant culture assays, they find that Nelfb promotes open chromatin structures and stabilizes RNA Pol II binding to a variety of adipogenic genes. Ectopic introduction of Pparg or treatment with a Pparg agonist restored adipocyte differentiation and prolonged lifespan of mutant mice. This work is succinct and shows how Nelfb controls Pparg expression to promote dWAT, an essential component of the skin that is necessary for postnatal development. Only minor clarifications are required.

- 1) Although Pparg agonist treatment to postnatal day 5 allowed mice to survive to 15 days, it is unclear if maintenance of dWAT is altered after treatment, leading to eventual death. Is dWAT reduced after cessation of agonist treatment or is a non-Pparg-dependent pathway important for survival?
- 2) Are any human conditions related to insufficient dWAT that agonist treatment might provide a therapy?

Reviewer 2

In this manuscript, Mahapatra et al. investigated the role of NELF, a transcriptional elongation and pausing complex, in dermal white adipose tissue (dWAT) development and adipocyte differentiation. They demonstrated that loss of Nelfb in Pdgfra+ preadipocyte lineages resulted in failure of dWAT formation, with impaired differentiation into mature adipocytes. Mechanistically, Nelfb supports chromatin accessibility and stabilizes RNA Polymerase II binding at key adipogenic loci such as Pparg, Cebpa to allow their proper transcription. Functional rescue using Pparg overexpression or rosiglitazone treatment supports its role in adipogenesis. While the data are generally convincing, several mechanistic aspects require clarification or further experimentation, as detailed below.

Major comments

1. While NELF is known to regulate transcriptional pausing/elongation, it remains unclear whether the phenotype observed upon Nelfb deletion is directly due to defects in this process. Rescue via Pparg overexpression could suggest a simpler transcriptional downregulation rather than a pausing/elongation-specific defect. To support this claim, the authors should consider ChIP analysis of RNA Polymerase II Ser2 phosphorylation at the Pparg locus during differentiation upon Nelfb deletion. Alternatively, administration of a Pol II elongation inhibitor could recapitulate the phenotypes observed in Nelfb-depleted cells or mice.
2. The observed decrease in H3K4me3/H3K27ac and increase in H3K27me3/H3K9me3 is intriguing. It remains unclear whether these epigenetic changes reflect a direct alteration in chromatin accessibility or are secondary consequences of reduced transcription. The authors should elaborate or experimentally address this point.

Minor comments

1. Given that NELFb loss naturally reduces its own binding, it would be helpful to assess binding of other NELF components by ChIP to evaluate complex-wide disruption if good antibodies are available.
2. The manuscript lacks a rationale for targeting Nelfb. Do other NELF subunits yield similar phenotypes when deleted? This distinction may influence whether the manuscript should emphasize NELFb-specific effects or refer to the broader NELF complex.
3. The selective enrichment of NELFb at lineage-determining genes deserves further explanation. Is this based on enhancer/promoter sequence features, chromatin state, or TF recruitment?
4. The upregulation of Wnt and IL-6 pathways in Nelfb-deficient cells could contribute to impaired differentiation. Testing whether inhibition of these pathways rescues adipogenesis would significantly strengthen the mechanistic conclusions.

First revision

Author response to reviewers' comments

We thank the reviewers for their suggestions that have made this manuscript substantially stronger. We have now addressed all of the reviewers' concerns via additional experiments, further analysis, or better clarification. Please see below for a point-by-point response to each reviewer's question. New text written in the manuscript as a response to reviewer questions are underlined.

Reviewer 1

We thank the Reviewer for helpful and positive comments on the paper, which we believe have provided valuable guidance in improving the paper.

1) Although Pparg agonist treatment to postnatal day 5 allowed mice to survive to 15 days, it is unclear if maintenance of dWAT is altered after treatment, leading to eventual death. Is dWAT reduced after cessation of agonist treatment or is a non-Pparg-dependent pathway important for survival?

Yes, dWAT is reduced upon cessation of Rosiglitazone treatment. This is now shown in Figure 5C-5D and stated in the results section as, "Interestingly, Oil Red O staining of the dorsal skin of Rosi treated *Nelfb*^{-/-} mice that survived to days 14 and 15 showed that these mice had lost their dWAT (Figure 5C-5D). This suggests that continued treatment with the Pparg agonist Rosi is essential for maintenance of mature adipocytes in *Nelfb* deleted mice (Figure 5C-5D)."

2) Are any human conditions related to insufficient dWAT that agonist treatment might provide a therapy?

Yes indeed, anti-EGFR therapy used to treat a variety of cancers can lead to papulopustular rashes with atrophy of dWAT. We have now included this in the Discussion section which is stated as, "Similar approaches have been used to treat the atrophy of dWAT due to anti-epidermal growth factor receptor (EGFR) therapy. EGFR is a therapeutic target for a variety of cancers however anti-EGFR therapy often leads to cutaneous toxicity characterized by papulopustular rashes with atrophy of dWAT. A recent study using a mouse model of anti-EGFR therapy showed that rosiglitazone treatment led to the expansion of dWAT which decreased the severity of rashes¹."

Reviewer 2

We thank the reviewer for the constructive comments that have made this manuscript significantly stronger.

1. While NELF is known to regulate transcriptional pausing/elongation, it remains unclear whether the phenotype observed upon *Nelfb* deletion is directly due to defects in this process. Rescue via *Pparg* overexpression could suggest a simpler transcriptional downregulation rather than a pausing/elongation-specific defect. To support this claim, the authors should consider ChIP analysis of RNA Polymerase II Ser2 phosphorylation at the *Pparg* locus during differentiation upon *Nelfb* deletion. Alternatively, administration of a Pol II elongation inhibitor could recapitulate the phenotypes observed in *Nelfb*-depleted cells or mice.

We agree with the reviewer and have now performed CUT&RUN using an RNA polymerase II Ser2 phosphorylation antibody. We have now shown that there is significant binding of RNA polymerase II Ser2 phosphorylation to *Pparg*, *Cebpa*, *Stat3*, and *Krox20* at the TSS and gene body regions of control but not *Nelfb* deleted cells (Figure 2I-J). This is now stated in the results section as,

"To see if *Nelfb* was directly influencing transcription elongation, CUT&RUN using an antibody against elongating forms of RNA Polymerase II (RNA Pol II Ser2 phosphorylation) was done. Robust RNA Pol II Ser2 phosphorylation could be detected on the TSS and gene body regions of *Pparg*, *Cebpa*, *Stat3*, and *Krox20* which disappeared upon deletion of *Nelfb* (Figure 2I-J). This suggests *Nelfb* is necessary for the elongation of these genes during adipogenic differentiation."

2. The observed decrease in H3K4me3/H3K27ac and increase in H3K27me3/H3K9me3 is intriguing. It remains unclear whether these epigenetic changes reflect a direct alteration in chromatin accessibility or are secondary consequences of reduced transcription. The authors should elaborate or experimentally address this point.

We thank the reviewer for this interesting question. The relationship between transcription and chromatin structure is an active area of investigation with many papers published solely focused on this question. Whether loss of chromatin accessibility is a direct consequence of reduced transcription is controversial and results dependent upon system used as well as timing of experiments. Two papers published in *Cell*² and *Nature*³ suggested that transcribing RNA polymerase II directly promotes open chromatin by displacing cohesin binding. They demonstrated that small molecule blockade of transcription leads to cohesin binding and chromatin compaction. A more recent paper published in *Genome Biology*⁴ used the auxin inducible degron system to acutely degrade RNA polymerase II and found that only upon long periods of RNA polymerase II depletion is there loss of chromatin accessibility and cohesin binding. Thus the authors of that study concluded that alterations in chromatin accessibility are secondary consequences of reduced transcription. Since these are major questions in the chromatin/transcription field that warrant extensive study as well as specialized techniques such as the auxin inducible degron system, we believe addressing this question experimentally is beyond the scope of this manuscript. However we have added additional explanation of this in the Discussion section of the manuscript which is stated as, “Nelfb loss potentially causes compacted chromatin due to the loss of RNA Pol II binding and thus loss of transcription. The process of RNA Pol II mediated transcription is essential for chromatin decompaction and loss of it leads to chromatin closure^{5,6}. Previous studies using small molecule inhibitors of RNA Pol II have shown that transcription directly promotes open chromatin through displacement of Cohesin binding^{2,3}. However, a study using acute degradation of RNA Pol II using the auxin inducible degron system showed that loss of transcription indirectly caused chromatin closure⁴. It is currently unclear whether the chromatin closure seen in the adipogenic genes due to Nelfb loss is a direct or indirect consequence of reduced transcription and would be an area of interest for future investigation.”

Minor comments

1. Given that NELFb loss naturally reduces its own binding, it would be helpful to assess binding of other NELF components by ChIP to evaluate complex-wide disruption if good antibodies are available.

We have now performed a Nelfe CUT&RUN in control and Nelfb deleted cells. Nelfe does not bind to *Pparg*, *Krox20*, or *Cebpa* in either control or Nelfb^{-/-} cells. Nelfe does bind to Stat3 in both control and Nelfb^{-/-} cells (Supplemental Figure 5C). This is now stated in the results section as, “To investigate whether other NELF complex members bind to the same adipogenic genes as Nelfb, CUT&RUN using an antibody against Nelfe was performed in control and Nelfb deleted cells. There was no significant binding of Nelfe to the TSS regions of *Pparg*, *Krox20*, or *Cebpa* (Figure S5C). However, there was binding of Nelfe to Stat3 in both control and Nelfb depleted cells. These data suggests that Nelfb specifically binds to the adipogenic transcription factors since other NELF complex members such as Nelfe don’t bind to most of them.”

2. The manuscript lacks a rationale for targeting Nelfb. Do other NELF subunits yield similar phenotypes when deleted? This distinction may influence whether the manuscript should emphasize NELFb-specific effects or refer to the broader NELF complex.

We have now tried this experiment by knocking down Nelfa and Nelfe. Loss of Nelfa or Nelfe slightly increased adipocyte differentiation gene expression (Figure S5). Thus there is a Nelfb specific requirement for adipogenesis. We have now changed all our statements from a broader NELF complex to Nelfb specific effects. We have also stated in the Results section, “To determine if control of adipocyte differentiation is applicable to all NELF complex members or Nelfb specific, Nelfa and Nelfe were knocked down using siRNAs. Interestingly, loss of Nelfa resulted in the slight upregulation of differentiation genes such as *Acot4*, *Plaat3*, *Elovl6*, *Pparg*, *Cebpa*, *Pln1*, *Fabp4* but downregulation of *Krox20* (Figure S5A). Knockdown of Nelfe led to slight

increases in the expression of *Acot4*, *Pparg*, *Cebpa*, and *Fabp4* but downregulation of *Adig* (Figure S5B). Depletion of *Nelfa* did not alter the expression of *Nelfe* nor did knockdown of *Nelfe* impact *Nelfa* levels (Figure S5A-S5B). Taken together these results suggest that *Nelfe* and *Nelfa* are not necessary to promote adipocyte differentiation. Furthermore, there is a *Nelfb* specific control of adipogenic differentiation.”

We have also added in the Discussion section the following statement,

“It is also worth noting that there is a *Nelfb* specific requirement for promoting adipocyte differentiation through RNA Pol II stabilization. *Nelfa* or *Nelfe* knockdown using siRNAs did not block differentiation gene expression but instead slightly increased it. In addition, *Nelfe* did not bind a majority of the genes that *Nelfb* binds and regulates. Other reports where deletion of *Nelfb* has been shown to have a functional role in T cells, muscle stem cells, and cardiomyocytes have all attributed the phenotype to the entire NELF complex^{30 28 26}. However the other subunits were not targeted in these studies and may be worth exploring to determine if the phenotypes were *Nelfb* specific or can be attributed to the broader NELF complex.”

3. The selective enrichment of NELFb at lineage-determining genes deserves further explanation. Is this based on enhancer/promoter sequence features, chromatin state, or TF recruitment?

We agree with the reviewer that *Nelfb* may potentially be recruited by transcription factors since it has been shown that TCF1 can recruit NELFB to its target genes in CD8+ T cells⁷. To determine if any transcription factor binding sites are enriched in the *Nelfb* bound lineage determining genes (*Pparg*, *Cebpa*, *Stat3*, *Krox20*) we used TRANSFAC¹⁰. Interestingly, the Sp1 transcription factor binding motif is the only one found to be present in all 4 genes (Figure S4F). Sp1 has been shown to be a repressor of *Cebpa* expression in preadipocytes by binding to its promoter region¹¹. Notably Sp1 expression is diminished upon differentiation which allows for transcription of *Cebpa*. Since *Nelfb* is already bound to the lineage determining genes in preadipocytes with paused RNA pol II (repressed state), Sp1 may potentially be the factor that recruits *Nelfb* to those sites (Figure S4C).

This is now stated in the results section as, “Since *Nelfb* is already bound to *Pparg*, *Cebpa*, and *Krox20* in progenitor cells, we wanted to determine if these genes were enriched in any transcription factor motifs that could potentially recruit *Nelfb* to these sites (Figure S4C). A search of the promoter region using TRANSFAC¹⁰ (-500bp to TSS) of *Pparg*, *Cebpa*, *Stat3*, and *Krox20* showed that only the SP1 binding motif was found in all 4 genes (Figure S4F). Sp1 has been previously shown to bind and repress *Cebpa* in adipocyte progenitor cells¹¹. Sp1 may potentially repress these genes by recruiting *Nelfb* to these sites and promoting RNA Pol II pausing and thereby preventing expression of these genes in preadipocytes.”

4. The upregulation of Wnt and IL-6 pathways in Nelfb-deficient cells could contribute to impaired differentiation. Testing whether inhibition of these pathways rescues adipogenesis would significantly strengthen the mechanistic conclusions.

We thank the reviewer for this question. The upregulation of Wnt and Il6 pathway genes are likely indirect targets of *Nelfb* and due to *Nelfb* deleted cells being unable to differentiate into adipocytes due to loss of expression of key adipogenic transcription factors such as *Pparg* and *Cebpa*. Since the *Nelfb* knockout cells can't differentiate, they are locked in the adipocyte progenitor state that are known to express higher levels of Wnt and Il6 pathway genes. This is now stated in the Discussion section as, “Wnt pathway genes are well-described inhibitors of adipogenesis and are highly expressed in progenitor cells to prevent differentiation^{12 13}. The upregulation of Wnt-related genes such as *Wnt10b*, *Sox9*, and *Dlk1* in *Nelfb*^{-/-} cells suggested that these cells are trapped in a progenitor state and are unable to differentiate. In support of *Nelfb* deleted cells being locked in the progenitor state, *IL6*, a lipolytic cytokine known to be expressed in preadipocytes^{14 15}, was also upregulated in *Nelfb*^{-/-} cells.”

In addition, we have already shown that *Nelfb* direct regulation through RNA polymerase II mediated transcription of adipogenic transcription factors is necessary for adipogenesis. In particular, *Pparg* expression through retroviruses can restore adipocyte differentiation in *Nelfb*

deleted cells (Figure 3). Rosiglitazone treatment of Nelfb deleted mice also led to restoration of dWAT (Figure 4).

1. Chen, L., You, Q., Liu, M., Li, S., Wu, Z., Hu, J., Ma, Y., Xia, L., Zhou, Y., Xu, N., and Zhang, S. (2022). Remodeling of dermal adipose tissue alleviates cutaneous toxicity induced by anti-EGFR therapy. *Elife* 11. 10.7554/eLife.72443.
2. Heinz, S., Texari, L., Hayes, M.G.B., Urbanowski, M., Chang, M.W., Givarkes, N., Rialdi, A., White, K.M., Albrecht, R.A., Pache, L., et al. (2018). Transcription Elongation Can Affect Genome 3D Structure. *Cell* 174, 1522-1536 e1522. 10.1016/j.cell.2018.07.047.
3. Busslinger, G.A., Stocsits, R.R., van der Lelij, P., Axelsson, E., Tedeschi, A., Galjart, N., and Peters, J.M. (2017). Cohesin is positioned in mammalian genomes by transcription, CTCF and Wapl. *Nature* 544, 503-507. 10.1038/nature22063.
4. Jiang, Y., Huang, J., Lun, K., Li, B., Zheng, H., Li, Y., Zhou, R., Duan, W., Wang, C., Feng, Y., et al. (2020). Genome-wide analyses of chromatin interactions after the loss of Pol I, Pol II, and Pol III. *Genome biology* 21, 158. 10.1186/s13059-020-02067-3.
5. Gu, B., Swigut, T., Spencley, A., Bauer, M.R., Chung, M., Meyer, T., and Wysocka, J. (2018). Transcription-coupled changes in nuclear mobility of mammalian cis-regulatory elements. *Science* 359, 1050-1055. 10.1126/science.aao3136.
6. Jubb, A.W., Boyle, S., Hume, D.A., and Bickmore, W.A. (2017). Glucocorticoid Receptor Binding Induces Rapid and Prolonged Large-Scale Chromatin Decompaction at Multiple Target Loci. *Cell Rep* 21, 3022-3031. 10.1016/j.celrep.2017.11.053.
7. Wu, B., Zhang, X., Chiang, H.C., Pan, H., Yuan, B., Mitra, P., Qi, L., Simonyan, H., Young, C.N., Yvon, E., et al. (2022). RNA polymerase II pausing factor NELF in CD8(+) T cells promotes antitumor immunity. *Nature communications* 13, 2155. 10.1038/s41467-022-29869-2.
8. Pan, H., Qin, K., Guo, Z., Ma, Y., April, C., Gao, X., Andrews, T.G., Bokov, A., Zhang, J., Chen, Y., et al. (2014). Negative elongation factor controls energy homeostasis in cardiomyocytes. *Cell Rep* 7, 79-85. 10.1016/j.celrep.2014.02.028.
9. Robinson, D.C.L., Ritso, M., Nelson, G.M., Mokhtari, Z., Nakka, K., Bandukwala, H., Goldman, S.R., Park, P.J., Mounier, R., Chazaud, B., et al. (2021). Negative elongation factor regulates muscle progenitor expansion for efficient myofiber repair and stem cell pool repopulation. *Dev Cell* 56, 1014-1029 e1017. 10.1016/j.devcel.2021.02.025.
10. Wingender, E., Chen, X., Hehl, R., Karas, H., Liebich, I., Matys, V., Meinhardt, T., Pruss, M., Reuter, I., and Schacherer, F. (2000). TRANSFAC: an integrated system for gene expression regulation. *Nucleic Acids Res* 28, 316-319. 10.1093/nar/28.1.316.
11. Tang, Q.Q., Jiang, M.S., and Lane, M.D. (1999). Repressive effect of Sp1 on the C/EBPalpha gene promoter: role in adipocyte differentiation. *Mol Cell Biol* 19, 4855-4865. 10.1128/MCB.19.7.4855.
12. Christodoulides, C., Lagathu, C., Sethi, J.K., and Vidal-Puig, A. (2009). Adipogenesis and WNT signalling. *Trends Endocrinol Metab* 20, 16-24. 10.1016/j.tem.2008.09.002.
13. Yang Loureiro, Z., Joyce, S., DeSouza, T., Solivan-Rivera, J., Desai, A., Skritakis, P., Yang, Q., Ziegler, R., Zhong, D., Nguyen, T.T., et al. (2023). Wnt signaling preserves progenitor cell multipotency during adipose tissue development. *Nat Metab* 5, 1014-1028. 10.1038/s42255-023-00813-y.
14. Radvanyi, A., and Roszer, T. (2024). Interleukin-6: An Under-Appreciated Inducer of Thermogenic Adipocyte Differentiation. *International journal of molecular sciences* 25. 10.3390/ijms25052810.
15. Harkins, J.M., Moustaid-Moussa, N., Chung, Y.J., Penner, K.M., Pestka, J.J., North, C.M., and Claycombe, K.J. (2004). Expression of interleukin-6 is greater in preadipocytes than in adipocytes of 3T3-L1 cells and C57BL/6J and ob/ob mice. *J Nutr* 134, 2673-2677. 10.1093/jn/134.10.2673.

Second decision letter

MS ID#: dev.204976R1

MS TITLE: Nelfb Promotes Dermal White Adipose Tissue Formation through RNA Polymerase II Mediated Adipogenic Gene Regulation

AUTHORS: George Sen; Samiksha Mahapatra; Julian Gomez; Uyanga Batzorig; Ye Liu; Celia Fernandez-Mendez; Yifang Chen

Dear Dr Sen,

I am happy to tell you that your manuscript has been accepted for publication in Development, pending our standard publication integrity checks.

Reviewer 1

The revised manuscript by Mahapatra et al. have addressed my concerns and is now suitable for publication.

Reviewer 2

It was surprising to see it function as a Nelfb rather than as a Nelf complex. The revised version addressed my and other reviewers' concerns.